# Brain clusterin protein isoforms and mitochondrial localization

Sarah K Herring[1], Hee-Jung Moon[1], Punam Rawal[1], Anindit Chhibber[1], Liqin Zhao[1,2]*

[1]Department of Pharmacology and Toxicology, School of Pharmacy, University of Kansas, Lawrence, United States; [2]Neuroscience Graduate Program, University of Kansas, Lawrence, United States

**Abstract** Clusterin (CLU), or apolipoprotein J (ApoJ), is the third most predominant genetic risk factor associated with late-onset Alzheimer's disease (LOAD). In this study, we use multiple rodent and human brain tissue and neural cell models to demonstrate that CLU is expressed as multiple isoforms that have distinct cellular or subcellular localizations in the brain. Of particular significance, we identify a non-glycosylated 45 kDa CLU isoform (mitoCLU) that is localized to the mitochondrial matrix and expressed in both rodent and human neurons and astrocytes. In addition, we show that rodent mitoCLU is translated from a non-canonical CUG (Leu) start site in Exon 3, a site that coincides with an AUG (Met) in human CLU. Last, we reveal that mitoCLU is present at the gene and protein level in the currently available CLU$^{-/-}$ mouse model. Collectively, these data provide foundational knowledge that is integral in elucidating the relationship between CLU and the development of LOAD.

DOI: https://doi.org/10.7554/eLife.48255.001

## Introduction

Late-onset Alzheimer's disease (LOAD) is the most common form of Alzheimer's disease (AD) representing more than 95% of all AD cases. The etiology of LOAD is currently unknown, but prevailing hypotheses suggest that it probably includes an intricate interaction of genetic and environmental factors (*Verheijen and Sleegers, 2018*; *Stoccoro and Coppedè, 2018*; *Berson et al., 2018*; *Nativio et al., 2018*). Of the dozens of genetic risk factors that have been associated with increased risk for LOAD, two members of the apolipoprotein family reside in the top three: apolipoprotein E (ApoE) and apolipoprotein J (ApoJ), also known as clusterin (CLU) (*Alzforum Networking, 2019*). While the risk conferred by ApoE has been extensively characterized, the identification of CLU as a genetic risk factor is relatively recent (*Harold et al., 2009*; *Lambert et al., 2009*). Despite this novelty, it is established that the CLU polymorphism rs11136000 confers an increased risk of developing LOAD that is independent of ApoE ε4 status. As nearly 36% of the Caucasian population carries this CLU risk genotype (*Bertram et al., 2007*; *Braskie et al., 2011*), a thorough understanding of the physiological properties and roles of CLU in the brain is crucial for understanding the pathophysiological impact of CLU in LOAD.

The CLU gene is located on the short arm of chromosome 8 (8p21.1) in humans (*Dietzsch et al., 1992*), chromosome 14 (14 D1; 14 34.36 cM) in mice (*The Jackson Laboratory, 2019*) and chromosome 15 (15p12) in rats (*National Center for Biotechnology Information, 2019a*). The CLU gene is organized into 11 exons (two exons are untranslated) of different sizes, spanning a region of 18,115 bp (human), 13,065 bp (mouse) or 39,246 bp (rat). Transcription of the CLU gene results in the production of at least three mRNA isoforms in humans and mice (NM_001831.3, NR_045494.1, and NR_038335.1 in humans; NM_013492.3, XM_006518504.3, and XM_006518503.3 in mice) and one mRNA transcript in rats (NM_053021.2). The translated protein isoforms share a high degree of

*For correspondence:
lzhao@ku.edu

Competing interests: The authors declare that no competing interests exist.

sequential homology across species, with the highest degree of similarity occurring in the disulfide bonding region located near Exon 5 (*Londou et al., 2008*; *Purrello et al., 1991*). Within Exons 1–2, the variance between transcripts is limited to Exon 1, with three Exon 1 variants existing: Exon 1A, Exon 1B, and Exon 1C (*Ling et al., 2012*). The Exon 1A-9-containing mRNA transcript [NCBI Reference # NM_001831.3 (human) and NM_013492.3 (murine)] is thought to be translated into the mature secreted CLU (mCLU) protein isoform. mCLU [70–75 kDa (dimerized) and 36–39 kDa (reduced)] is initially translated from Exon 2 into a 448 (mouse) or 449 (human) amino acid (aa) pre-protein, which contains an N-terminal 22-amino-acid endoplasmic reticulum (ER)-signaling peptide (SP; located in Exon 2) and two nuclear localization sequences (NLS; located in Exon 3 and Exon 8). Following translation, the mCLU pre-protein is immediately targeted to the ER, where the SP is removed and the protein is glycosylated at six N-linked glycosylation sites (*Kapron et al., 1997*; *Jordan-Starck, 1994*). The intermediate glycoprotein is then directed to the Golgi body, where glycan trimming and additional modifications occur. The resultant glycoprotein is then cleaved to form the two glycosylated mCLU subunits: the alpha subunit [mCLUα, 34–37 kDa (mice) or 40 kDa (human)] and the beta subunit [mCLUβ, 36–39 kDa (mice) or 40 kDa (human)]. Prior to secretion, mCLUα and mCLUβ are disulfide-bonded to form the antiparallel, heterodimeric glycoprotein commonly referred to as mCLU (*Burkey et al., 1991*).

In addition to mCLU, several 'minor' or alternative protein isoforms have been reported. The primarily discussed alternative isoform, generally referred to as nuclear or intracellular CLU (nCLU or icCLU) (*Yang et al., 2000*; *Caccamo et al., 2003*; *Debure et al., 2003*; *Leskov et al., 2003*; *Kim et al., 2012a*), lacks Exon 2 and is translated from an in-frame methionine (Met) in Exon 3, which results in the generation of a CLU protein isoform that has an approximate molecular weight of 45–50 kDa (primarily detected in human cells in the cytosol and nucleus [*Leskov et al., 2003*; *Prochnow et al., 2013*]). However, the data concerning the physiological role of the nCLU/icCLU protein isoform(s) are less clear. The predominant hypothesis is that the 45–50 kDa protein isoform is solely pro-apoptotic in nature (*Yang et al., 2000*; *Debure et al., 2003*; *Kim et al., 2012a*; *Dia and Mejia, 2010*; *Kim et al., 2012b*). In support of this hypothesis, studies have demonstrated that mCLU and nCLU regulate certain cellular processes in opposite manners. For example, mCLU promotes cellular survival in human osteosarcoma cells by interacting with and stabilizing the Ku70/Bax complex (*Trougakos et al., 2009*). By contrast, overexpression of the proposed nCLU protein isoform is associated with significantly reduced cell growth (*Yang et al., 2000*) and increased anoikis (*Caccamo et al., 2003*), and has been shown to induce cell-cycle arrest and caspase-dependent apoptosis (*Scaltriti et al., 2004*). In addition, the data concerning the subcellular distribution and/or localization of CLU are, at best, controversial. Widespread, robust expression of CLU mRNA is detectable in the postnatal mouse brain beginning at postnatal day 7, with predominant CLU immunoreactivity localized to astrocytes (*Charnay et al., 2008*). Parallel to these findings, another study demonstrates prominent astrocytic CLU expression with no detectable CLU mRNA in oligodendrocytes, microglia, or neurons (*Morgan et al., 1995*). By contrast, an examination of CLU immunoreactivity during neuronal development indicates prominent CLU immunoreactivity in both cortical and hippocampal neurons in postnatal day 6 mouse pups, suggesting an important role for CLU in neuronal development (*O'Bryan et al., 1993*). Moreover, inconsistencies in the understanding of CLU neurophysiology still exist, most predominantly relating to the interplay between neurotoxic stimuli and CLU expression. For instance, CLU expression is significantly altered in the brain following traumatic brain injury (*Troakes et al., 2017*), ischemic stroke (*Han et al., 2001*; *Imhof et al., 2006*), drug-induced seizures (*Schreiber et al., 1993*), and ethanol exposure (*Kim et al., 2012a*; *Trindade et al., 2016*). However, these studies, which demonstrate the stress-mediated response of CLU, provide conflicting evidence as to whether CLU expression ameliorates or exacerbates cellular stress.

The aforementioned research gaps exist primarily because the majority of brain-based CLU research lacks an in-depth characterization of the CLU mRNA transcripts and protein isoform expression and distribution in a healthy un-challenged brain. Moreover, although CLU is sequentially homologous across species, the similarities between the rodent and human brain CLU protein expression profiles have not been thoroughly addressed. Thus, the primary goal of this study was to bring clarity to the field of brain-based CLU research by performing a thorough examination of the CLU expression profile in the rodent brain, in primary cultures of rodent and human neurons and astrocytes, and in rodent and human brain-derived cell lines. In doing so, we have identified the mRNA transcripts present in the rodent brain and the corresponding protein isoforms

that are translated from each transcript. We have also characterized the cellular and subcellular distribution of individual CLU protein isoforms in the whole rodent brain, as well as the cell-type and organelle-specific location of each isoform. Furthermore, we have demonstrated that the previously detected Exon-2-lacking CLU protein isoform is a mitochondrial CLU protein isoform (thus deemed mitoCLU), and that this isoform is 1) present in both neurons and astrocytes of rodent and human origins, 2) present in the mitochondrial matrix of healthy rodent brain, 3) translated from an mRNA transcript containing Exons 3–9, and 4) translated from a CUG start codon in rodents and from a canonical AUG start site in humans: all of which are novel findings in the field of CLU research. Last, we reveal for the first time that mitoCLU is present in the commercially available CLU$^{-/-}$ mouse model that was developed to mimic total CLU deficiency. Collectively, the data presented herein provide a complete picture of CLU gene transcription and of protein isoform expression and distribution in the healthy rodent brain, prove that the rodent brain may be utilized to study human brain-based CLU physiology and pathophysiology, and highlight a novel avenue of mitochondria-focused CLU research.

## Results

### Multiple CLU protein isoforms are robustly expressed in rodent brain tissue

Previous studies have indicated the expression of CLU mRNA in multiple tissue types, with mRNA predominantly detected in the central nervous system (CNS) (*Connor et al., 2001*; *de Silva et al., 1990a*). However, robust mRNA levels do not always translate to robust protein levels. Therefore, CLU protein expression was examined in a panel of rodent (male and female) tissues that were known to contain CLU mRNA (*de Silva et al., 1990a*). The data indicate the expression of four major CLU immunoreactive bands in a whole-brain lysate [CLU_49 kDa, CLU_45 kDa, CLU_39 kDa, and CLU_36 kDa (*Figure 1A*)], an expression pattern that was consistent throughout multiple brain regions [cortex, hippocampus, hypothalamus, and cerebellum; (Supplementary Figure A)]. Consistent with the previous literature, several CLU immunoreactive bands were also detected in peripheral tissues. However, because the primary focus of this study is brain CLU, these tissues were not investigated further in this study. As LOAD predominantly affects the cortex and hippocampus and is more prevalent in the female population, cortical and hippocampal tissues from female animals were utilized for the remainder of this study (where applicable).

### Brain CLU protein isoforms exhibit distinct subcellular localizations

Previous literature indicates the presence of CLU protein isoforms in multiple subcellular compartments; therefore, the subcellular distribution of brain CLU protein isoforms was also investigated. The data indicate the expression of two CLU immunoreactive bands in the cytosolic fraction (CLU_39 kDa and CLU_36 kDa), five CLU immunoreactive bands in the organelle fraction (CLU_60 kDa, CLU_49 kDa, CLU_45 kDa, CLU_39 kDa, and CLU_36 kDa), and one CLU immunoreactive band that is specific to the nuclear fraction (CLU_68 kDa) of cortical tissue (*Figure 1B*). Although three additional CLU immunoreactive bands were detected in the nuclear fraction (CLU_60 kDa, CLU_39 kDa, and CLU_36 kDa), analysis with fraction-specific markers indicates that these bands are probably the result of minor cytosolic (CLU_39 kDa and CLU_36 kDa) and organelle (CLU_60 kDa) contamination in the nuclear fraction (*Figure 1B*, lower panel). Consistent with our biochemical analyses, CLU immunoreactivity was detected in both the nuclear and the cytosolic compartments of neurons (*Figure 1C* and *Figure 1—figure supplement 1B*) with low CLU immunoreactivity present in GFAP-positive astrocytes (*Figure 1—figure supplement 1C*). Collectively, these data indicate that CLU protein isoforms are targeted to at least three distinct subcellular compartments (cytosol, ER/mitochondria, and nucleus) within rodent brain cells.

### Neurons and astrocytes exhibit distinct and overlapping CLU mRNA and protein profiles

To better understand the total CLU expression profile in differing brain cell types, CLU gene and protein expression was examined in cortical neurons and astrocytes. Exon 3–4, Exon 2 and Exon 1B mRNA was detected in both neurons and astrocytes, indicating the presence of at least one CLU

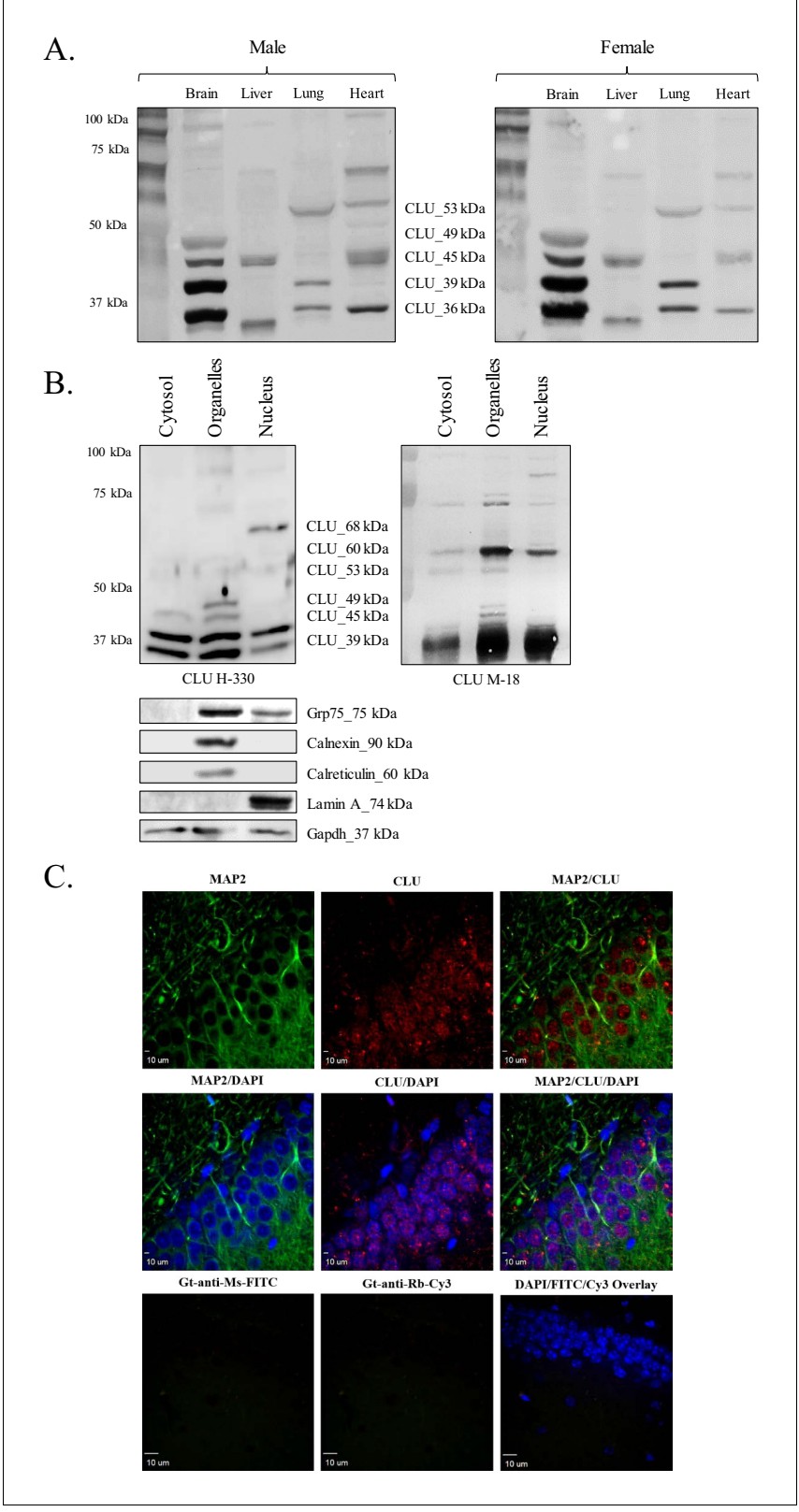

**Figure 1.** CLU protein is expressed in the brain as multiple protein isoforms that have distinct cellular localizations. (**A**) Whole brain, liver, lung, and heart tissues were isolated from age-matched male (left panel) and female (right panel) wild-type (WT) mice and homogenized as indicated. Total protein was analyzed with SDS-PAGE and blots were probed for CLU immunoreactivity with anti-CLU H-330. (**B**) Cytosolic, organelle, and nuclear fractions were

*Figure 1 continued on next page*

*Figure 1 continued*

isolated from freshly harvested WT cortical tissue as indicated. Fractions were analyzed via SDS-PAGE and blots were probed for CLU immunoreactivity with anti-CLU H-330 (left panel) and anti-CLU M-18 (right panel). Blots were stripped and re-probed with fraction-/organelle-specific biochemical markers: Grp75 (mitochondria), calnexin and calreticulin (ER), lamin A (nucleus), and Gapdh (cytosol). (**C**) 40-μM-thick rodent brain sections were permeabilized and blocked as indicated and labeled with anti-MAP2 (green) or anti-GFAP (*Figure 1—figure supplement 1C*) and anti-CLU H-330 (red). Brain sections were then washed and probed with anti-mouse FITC (for MAP2) or anti-rat Cy5 (for GFAP) and pre-adsorbed anti-rabbit Cy3 (for CLU). To generate a secondary antibody control (bottom panel), one group of free-floating brain sections was incubated overnight in the same conditions without primary antibody. Brain sections from the hippocampal dentate gyrus were imaged at 4X (*Figure 1—figure supplement 1B*) and 40X using a customized Olympus IX81/spinning disk confocal inverted microscope and analyzed using the Slidebook Software.

DOI: https://doi.org/10.7554/eLife.48255.002

The following figure supplement is available for figure 1:

**Figure supplement 1.** CLU protein expression in the rodent brain.

DOI: https://doi.org/10.7554/eLife.48255.003

mRNA transcript in each cell type. By contrast, Exon 1A and Exon 1C appeared to be expressed specifically in astrocytes and neurons, respectively (*Figure 2A*; right panel). These data suggest that both cell types are capable of translating CLU protein and that neuron- or astrocyte-specific expression profiles may exist. To investigate the CLU protein expression profile, cytosolic and nuclear fractions were harvested from cortical neurons or astrocytes and analyzed for CLU immunoreactivity. Analysis of the neuronal fractions indicates the expression of five CLU immunoreactive bands in the cytosolic fraction (CLU_60 kDa, CLU_53 kDa, CLU_49 kDa, CLU_45 kDa and CLU_39 kDa) and one CLU immunoreactive band in the nuclear fraction (CLU_68 kDa; *Figure 2B*, left panel). Analysis of astrocytic fractions indicates the expression of four CLU immunoreactive bands in the cytosolic fraction (CLU_60 kDa, CLU_49 kDa, CLU_45 kDa, and CLU_39 kDa) and one CLU immunoreactive band in the nuclear fraction (CLU_60 kDa; *Figure 2B*; right panel). Although several consistencies were noted between the neuron and astrocyte CLU protein expression profiles (CLU_60 kDa, CLU_49 kDa, CLU_45 kDa, and CLU_39 kDa), two CLU protein isoforms were observed exclusively in neurons (CLU_53 kDa and CLU_68 kDa). Collectively, these data confirm the presence of multiple CLU mRNA and protein isoforms in primary cultures of rodent neurons and astrocytes. In addition, as the neuronal CLU expression profile indicates neuron-specific expression of Exon 1C mRNA (*Figure 2A*; right panel) and of a CLU_53 kDa/CLU_68 kDa protein (*Figure 2B*; left panel), these data suggest that CLU_53 kDa and CLU_68 kDa protein isoforms are probably derived from an Exon 1C-containing CLU mRNA transcript.

## Identification of a mitochondrial CLU protein isoform

Preliminary analyses of CLU immunoreactivity in cortical neurons indicated diffuse punctate CLU staining that appeared to overlap with Mitotracker staining (*Figure 3A*). As multiple CLU protein isoforms were found to be localized to the organelle fraction of adult cortical tissue (*Figure 1B*), and as previous literature suggests the presence of a mitochondria-associated CLU protein isoform (*Debure et al., 2003*; *Trougakos et al., 2009*; *Zhang et al., 2005*), the question of a brain mitochondria-localized CLU protein isoform arises. Therefore, pure mitochondria were isolated from female cortical tissues and analyzed via immunoblotting. As expected, CLU_49 kDa, CLU_45 kDa, CLU_36–39 kDa, and CLU_60 kDa were detected in a whole brain lysate and the crude mitochondrial extract. By contrast, only the CLU_45-kDa protein isoform was detected in pure brain mitochondria (*Figure 3B*), indicating that CLU_45 kDa (deemed mitoCLU) is localized to rodent brain mitochondria under physiological conditions, a novel finding in the field of brain-based CLU research. To ensure that mitoCLU is not the result of an antibody artifact, mouse neuroblastoma cells (Neuro-2a), which robustly express the 45-kDa protein isoform in a manner that is similar to rodent brain tissue (*Figure 4A*), were transfected with scramble siRNA or CLU Exon 3-targeting siRNA. Immunoblot analysis indicated an approximate 60–65% decrease in CLU_45 kDa expression (*Figure 4B*) that was not observed in the scramble siRNA-transfected group. Although not analyzed statistically, these data indicate that the observed immunoreactive band is not the result of antibody artifact.

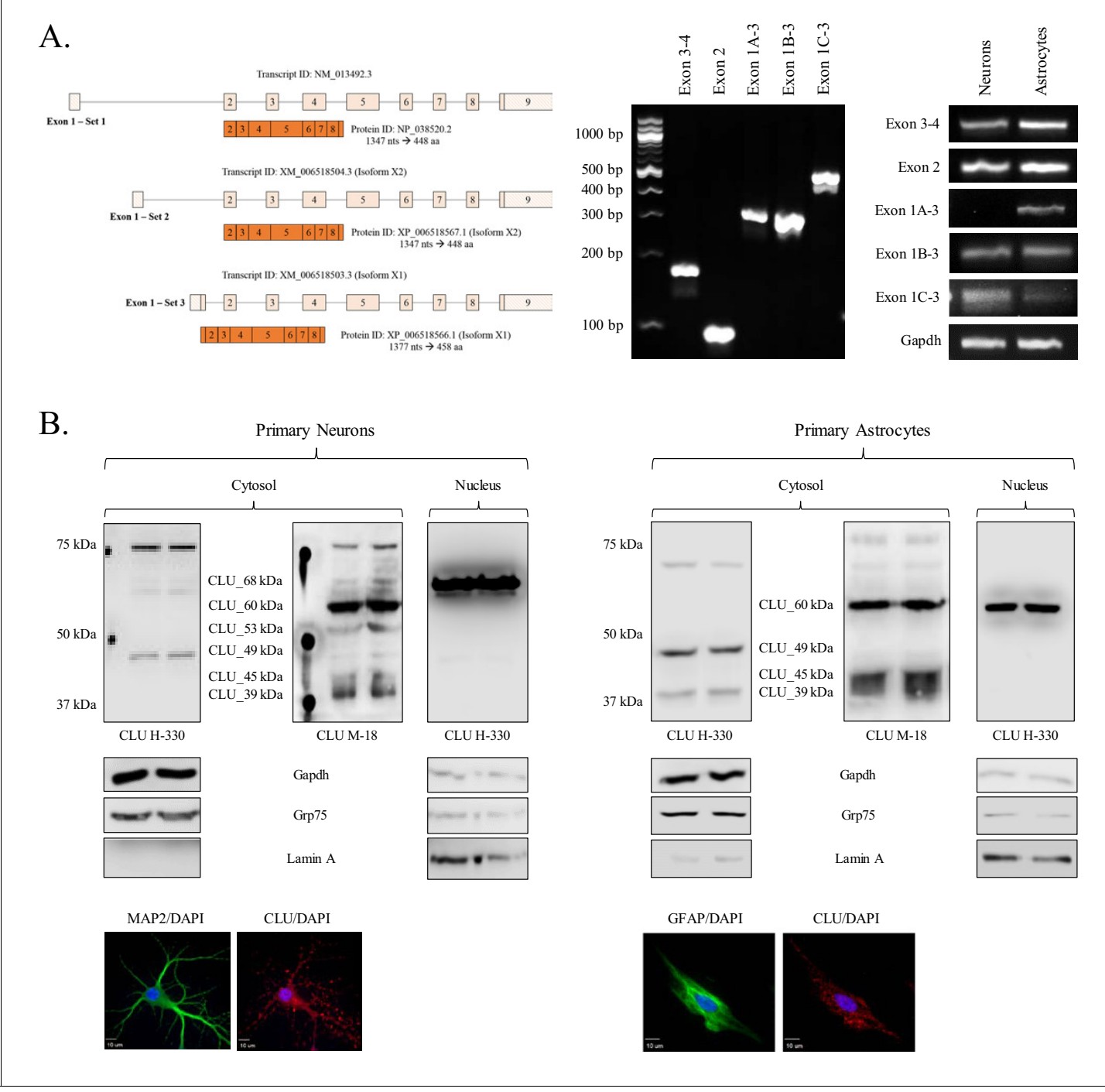

**Figure 2.** Characterization of neuronal and astrocytic CLU protein isoforms. (**A:** left panel) Schematic of known murine CLU mRNA transcripts provided in the NCBI database. (**A:** right panel) Specific CLU gene transcript levels were examined using 25 ng cDNA isolated from DIV 9/16 neurons/astrocytes. Amplicons were visualized on a 4% agarose gel to ensure the appropriate size (left panel) and a comparison of CLU amplicon intensity was visualized (right panel). (**B**) Primary neurons (left panel) or astrocytes (right panel) were prepared as indicated. At DIV 9/16, cytosolic and nuclear fractions were isolated. 30 μg of each fraction was analyzed for CLU protein expression using anti-CLU H-330 and anti-CLU M-18. Fraction purity was analyzed using Gapdh (cytosol), Grp75 (mitochondria), and lamin A (nucleus). Isolation of cell type was confirmed by double labeling with either MAP2 (neurons) or GFAP (astrocytes) and CLU H-330, as indicated in the Materials and methods. Cells were imaged at 40X (air; neurons) and 60X (oil; astrocytes) using a customized Olympus IX81/spinning disk confocal inverted microscope and analyzed using the Slidebook Software.

DOI: https://doi.org/10.7554/eLife.48255.004

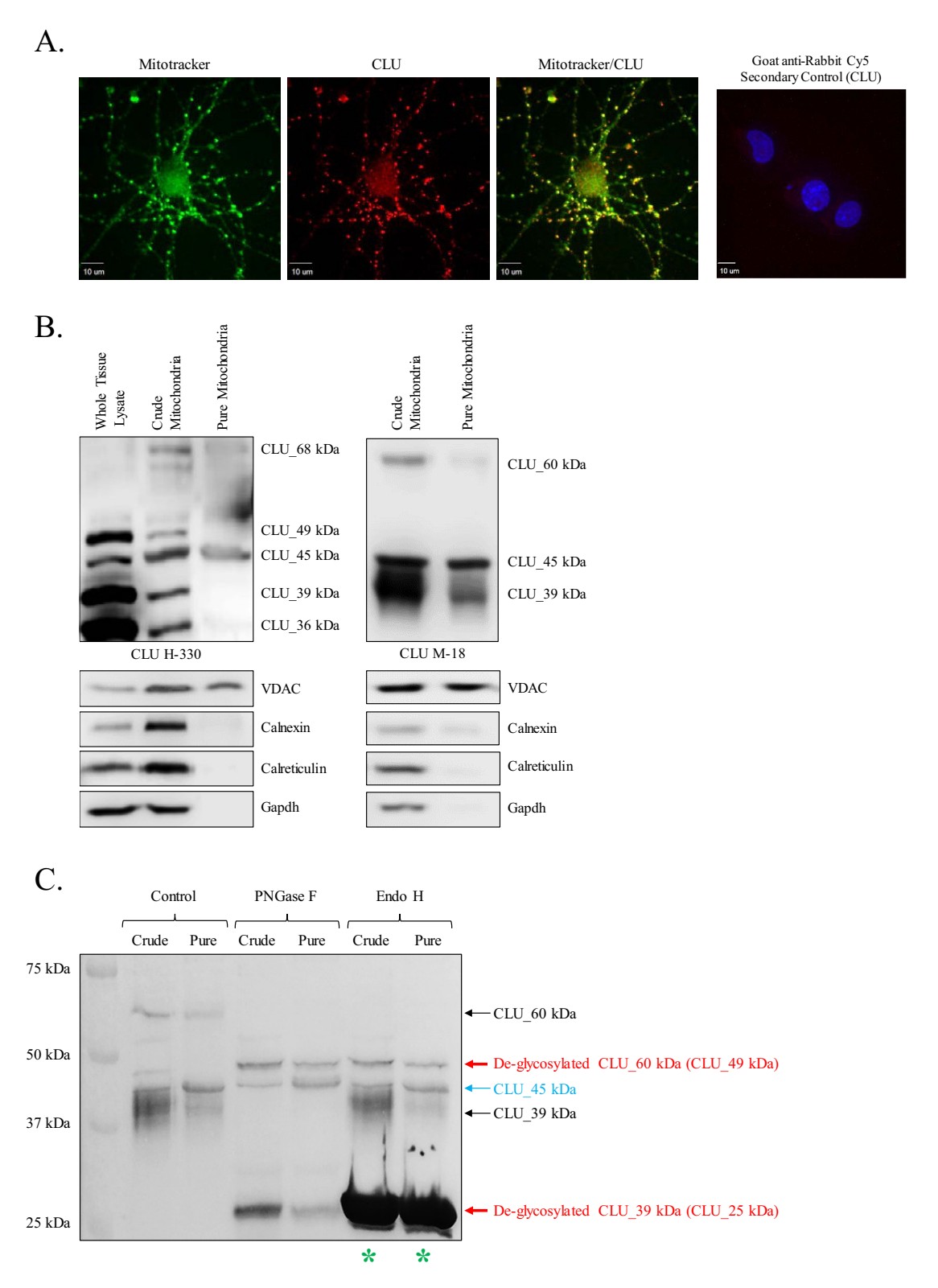

**Figure 3.** Identification of a mitochondrial CLU protein isoform. (**A**) DIV 9 Mitotracker-stained (green) primary neurons were probed for CLU immunoreactivity using anti-CLU H-330 (red) and visualized using 40X confocal microscopy. (**B**) Pure cortical mitochondria were isolated as indicated. Equal concentrations of whole tissue lysate, crude mitochondria, and pure mitochondria were analyzed via SDS-PAGE and probed for CLU immunoreactivity using anti-CLU H-330 (left panel) and anti-CLU M-18 (right panel) (n = 3 independent isolations). Biochemical characterization of
*Figure 3 continued on next page*

*Figure 3 continued*

isolated fractions was performed using a panel of organelle-specific antibodies: voltage-dependent anion channels (VDAC) (mitochondria), calnexin and calreticulin (ER), and Gapdh (cytosol). (**C**) Crude and pure mitochondria were isolated and subjected to endoglycosidase treatment using PNGase F and Endo H. Deglycosylated mitochondrial lysates were then analyzed for CLU immunoreactivity using anti-CLU M-18. Red font: deglycosylated protein isoforms; blue font: isoforms that were unaffected by glycosidase treatment; green asterisk: excess Endo H enzyme.

DOI: https://doi.org/10.7554/eLife.48255.005

The following figure supplement is available for figure 3:

**Figure supplement 1.** Positive controls for deglycosylation studies.

DOI: https://doi.org/10.7554/eLife.48255.006

## Mitochondrial CLU (CLU_45 kDa) is non-glycosylated

N-linked glycosylation is a major modifier of mature CLU, contributing approximately 30% of the final protein mass (*de Silva et al., 1990b*). To determine whether mitoCLU is modified by N-linked glycosylation, crude and pure mitochondria were treated with endoglycosidases specifically targeting N-linked glycans and analyzed by immunoblot analysis. Deglycosylation of CLU_60 kDa by both PNGase F and Endo H indicates that CLU_60 kDa is a high mannose-modified form of CLU_49 kDa. Deglycosylation of CLU_39 kDa to CLU_25 kDa by PNGase F but not Endo H indicates that this CLU immunoreactive band is modified by sialic-acid-containing complex glycans; a finding that is consistent with previous studies. No alteration in the molecular weight (MW) or intensity of mitoCLU was

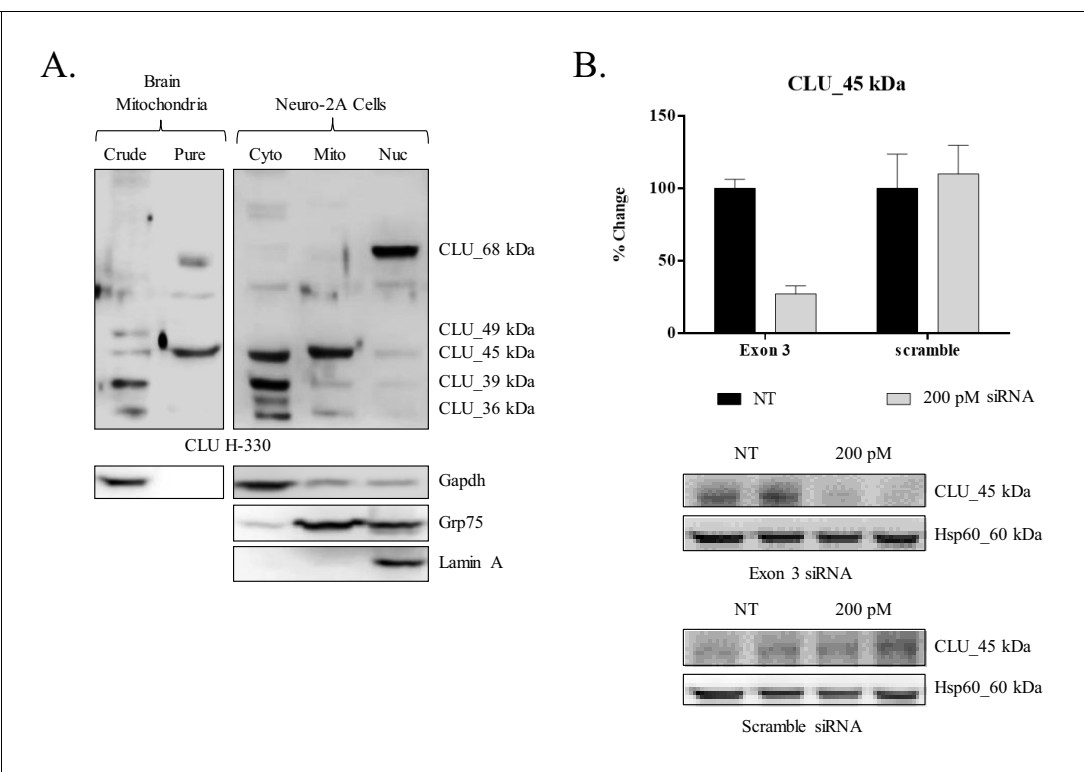

**Figure 4.** siRNA-mediated knockdown of CLU_45 kDa in mouse neuroblastoma cells. (**A**) Mouse neuroblastoma cells (Neuro-2a) were cultured and subjected to a subcellular fractionation to isolate the cytosolic, crude mitochondrial, and nuclear fractions. 30 μg of each fraction was analyzed for CLU protein expression using anti-CLU H-330. Fraction purity was analyzed using fraction-specific markers: Gapdh (cytosol), Grp75 (mitochondria), and lamin A (nucleus). (**B**) Neuro-2a cells were transfected with 200 pM of Exon-3-targeting siRNA or non-specific scramble for 18 hr followed by 72 hr incubation. Whole cell lysates were analyzed for CLU protein expression using anti-CLU H-330. To ensure consistent loading, blots were stripped and re-probed for Hsp60 immunoreactivity. The relative intensity of CLU_45 kDa was normalized to the relative intensity of Hsp60 and is plotted as % change compared to the non-transfected cells. Values represent the mean of two individual experiments, with error bars representing the standard error.

DOI: https://doi.org/10.7554/eLife.48255.007

noted following endoglycosidase treatment, indicating that mitoCLU is not a target for N-linked glycosylation or a precursor for a higher MW glycoprotein (*Figure 3C*).

## CLU_45 kDa is localized to the mitochondrial matrix of rodent brain

To determine the sub-mitochondrial localization of mitoCLU, cortical mitochondria were isolated and subjected to a mitochondria sub-fractionation. The data indicate prominent CLU_45 kDa immunoreactivity in the control group with a decrease of approximately 32% in CLU_45 kDa expression occurring between the control-treated group and the 0.1% digitonin-treated group. CLU_45 kDa immunoreactivity decreases approximately 8% between the 0.1%- and 0.2%-treated groups, but in the 0.3%-treated group, there is a sharp decline that results in the loss of 94% of mitochondrial CLU_45 kDa expression (*Figure 4B*; black line). This expression profile change does not match that of the voltage-dependent anion channels (VDAC) (*Figure 5B*; red line) or cytochrome C (Cyt. C) (*Figure 4B*; green line), indicating that CLU_45 kDa is not localized to the outer mitochondrial membrane (OMM) or intermembrane space (IMS). By contrast, the data indicate a certain degree of similarity between CLU_45 kDa and CoxIV (*Figure 4B*; purple line) expression changes. Specifically, the

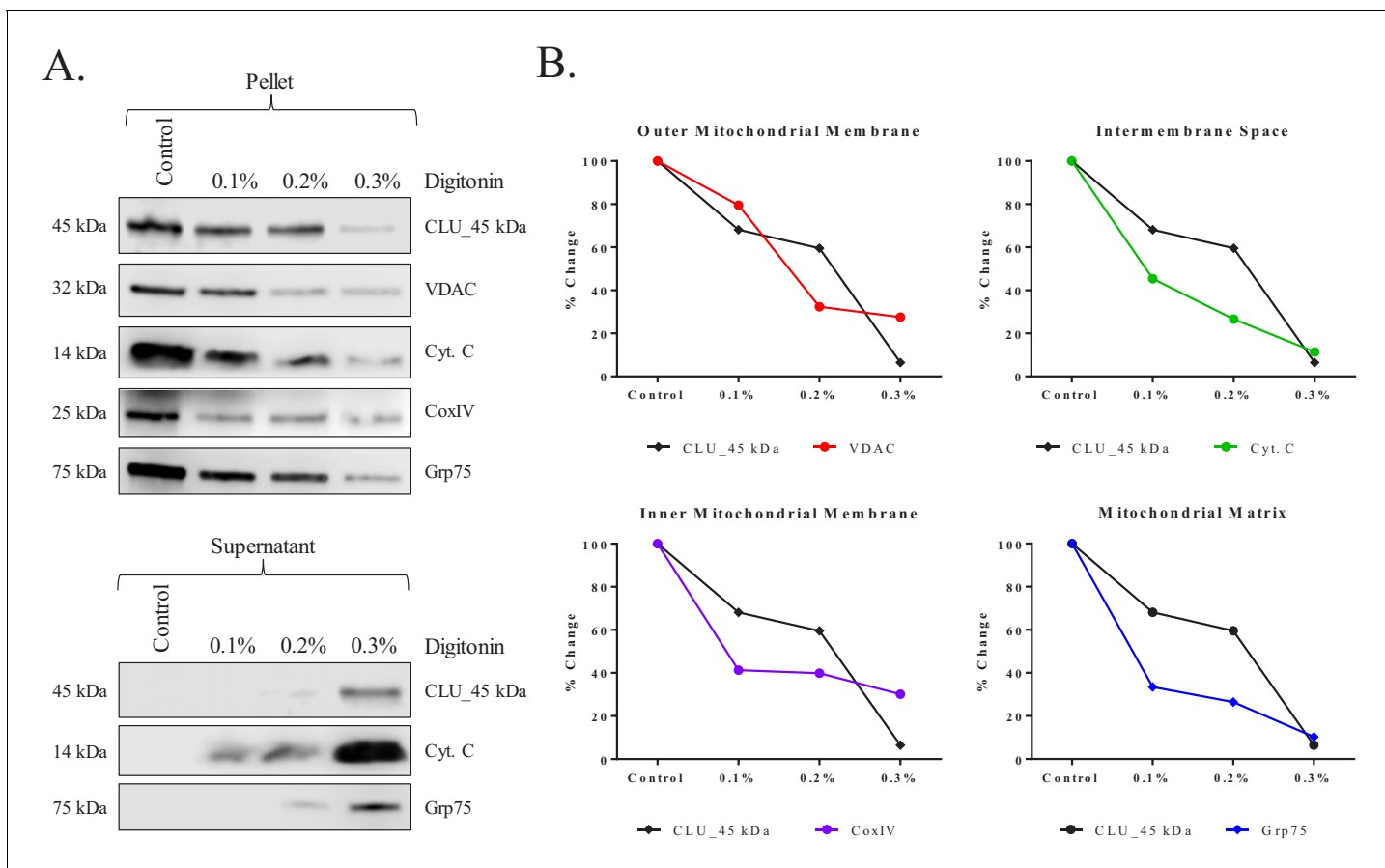

**Figure 5.** Submitochondrial localization of CLU_45 kDa in rodent brain mitochondria. Pure cortical mitochondria were isolated using a discontinuous percoll gradient and subjected to a mitochondrial subfractionation using increasing concentrations of digitonin (0%–0.3%). Following permeabilization, mitochondrial samples were separated to yield the mitochondrial pellet (retained proteins) and the mitochondrial supernatant (released proteins). (**A**) Samples were probed for CLU immunoreactivity with CLU H-330 and mitochondrial-compartment-specific markers: VDAC (outer mitochondrial membrane), cytochrome C (Cyt. C; intermembrane space), CoxIV (inner mitochondrial membrane), and Grp75 (Matrix). (**B**) Percent change was calculated by setting each protein control sample to 100%. CLU expression changes in the pellet were compared to the expression changes in each fraction-specific marker. In the provided plots, the y-axis represents % change in protein expression and the x-axis represents % digitonin from 0% (Control) to 0.3%. The black lines in each of the four plots represent CLU % change while the colored lines represent the % change of the respective mitochondrial marker.
DOI: https://doi.org/10.7554/eLife.48255.008

expression of CLU_45 kDa and CoxIV decline sharply between the control group and the 0.1%-treated group, with relatively little change occurring between the 0.1%- and 0.2%-treated groups. However, unlike CLU_45 kDa expression, which decreases by 54% between the 0.2%- and 0.3%-treated groups, the addition of 0.3% digitonin results in only a 10% decrease in CoxIV immunoreactivity when compared to the 0.2%-treated group. This suggests that CLU_45 kDa is localized to the mitoplast but not specifically to the inner mitochondrial membrane (IMM). Consistent with CoxIV, Grp75 (*Figure 4B*; blue line) immunoreactivity decreases sharply between the control and the 0.1%-treated groups, with only a slight decrease occurring between the 0.1%- and 0.2%-treated groups. However, unlike CoxIV expression, the addition of 0.3% digitonin results in the loss of nearly all Grp75 immunoreactivity, yielding an expression pattern that is nearly identical to that of CLU_45 kDa. These data suggest that CLU_45 kDa is localized to the mitochondrial matrix.

Consistent with these data, analyses of the supernatant fractions (representing the released proteins) indicated a lack of CLU_45 kDa or Grp75 immunoreactivity in the control and the 0.1% digitonin-treated groups, with only minute protein immunoreactivity in the 0.2%-treated group. However, a robust increase in the presence of both proteins is observed in the 0.3%-treated supernatants, further suggesting that CLU_45 kDa is released from the mitochondrial matrix in a manner that is similar to that of soluble matrix proteins (*Figure 5A*; lower panel). Collectively, these data indicate that the CLU_45 kDa protein isoform is localized to the mitochondrial matrix of rodent brain mitochondria. To our knowledge, this is the first report to indicate the localization of an intracellular CLU protein isoform to the matrix of brain mitochondria.

## Murine CLU_45 kDa is translated from a CUG (Leu) start site in Exon 3

Prior deglycosylation studies indicate that CLU_45 kDa is not glycosylated (*Figure 3C*) and therefore does not contain Exon 2. Therefore, it was presumed that translation of CLU_45 kDa was initiated in Exon 3. However, sequential analysis and alignment of human and murine CLU indicates that Exon 3 of murine CLU does not contain an ATG (*Figure 6A*). Instead, murine Exon 3 contains a CTG, which encodes a Leu that is capable of functioning as a non-canonical translational start site (*Kearse and Wilusz, 2017*). To determine whether CLU_45 kDa is translated from this CTG, Myc-tagged Exon 2–9-containing or Exon 3–9-containing murine CLU constructs were overexpressed in Neuro-2a cells, and cell lysates (*Figure 6B*; upper panel) or culture media (*Figure 6B*; lower panel) were analyzed for CLU immunoreactivity. Overexpression of the Exon 2–9 construct yields mCLU protein isoforms that were previously detected in vivo and in primary cultures of neurons and astrocytes (CLU_60 kDa, CLU_49 kDa, and CLU_39 kDa). In support of these data, CLUα was detected in cell culture medium from cells that were transfected with only the Exon 2–9 construct (*Figure 6B*; lower panel). By contrast, overexpression of the Exon 3–9 construct resulted in the generation of intracellular CLU_45 kDa and two previously unobserved lower MW bands (below 39 kDa). These two bands correspond to the MW of protein isoforms translated from Exon 4 where, at least, two in-frame Met-encoding nucleotide triplets are located. Collectively, these data indicate that mitoCLU is translated from a non-canonical CUG (Leu) start site located in Exon 3.

## Overexpressed murine CLU_45 kDa is non-glycosylated

As overexpression of CLU constructs appears to replicate the in vivo CLU protein expression profile, deglycosylation studies were performed to ensure that the proposed Exon-3–9-derived isoform was not glycosylated. Consistent with our previous findings (*Figure 3C*), deglycosylation of CLU_60 kDa and CLU_39 kDa was observed in cells overexpressing Exon 2–9 following treatment with PNGase F and/or Endo H (*Figure 6C*; upper panel). As expected, secreted mCLU, from cells overexpressing Exon 2–9, was also deglycosylated (*Figure 6C*; lower panel). The analysis of lysates from cells overexpressing Exon 3–9 indicated that mitoCLU is not impacted by endoglycosidase treatment: a finding that is consistent with in vivo data from both crude and pure rodent brain mitochondria (*Figure 6C*; upper panel and *Figure 3C*). These data indicate that the in vivo CLU protein expression profile was successfully mimicked in transfected cells.

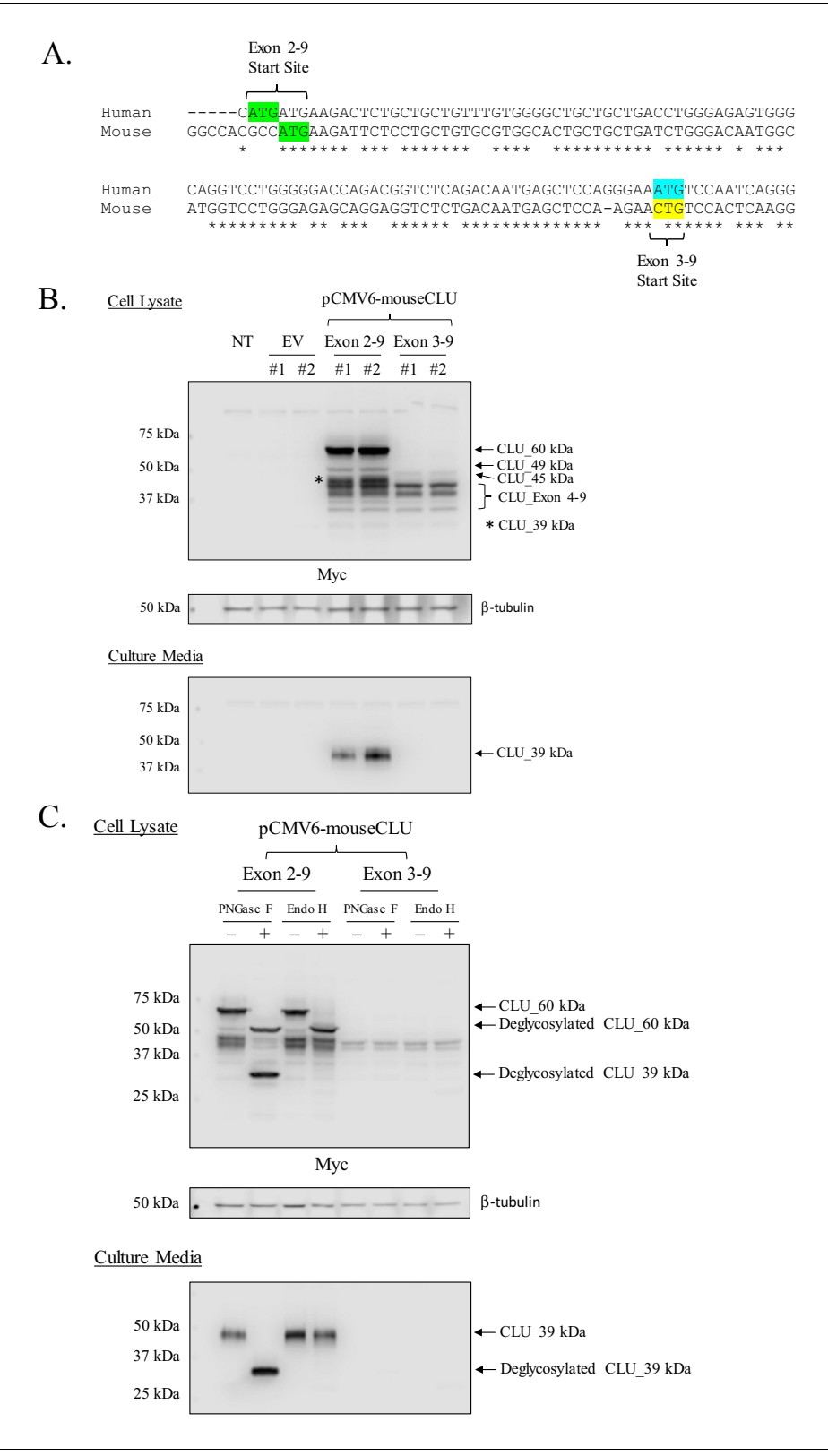

**Figure 6.** Mouse CLU_45 kDa is translated from a non-canonical CUG start site in Exon 3. (A) Sequential alignment of human and mouse CLU: DNA of Exons 2 and 3. Green sequences represent the Exon 2 translational start site (ATG) in both mouse and human CLU. Blue and yellow sequences represent the Exon 3 translational start site for human CLU (ATG) and the proposed Exon 3 translational start site for mouse CLU_45 kDa (CTG), respectively. (B) Empty vector (pCMV6), Myc-tagged pCMV6-mouse CLU Exon 2–9, or Exon 3–9 constructs were transfected into Neuro-2a cells. Cell

*Figure 6 continued on next page*

*Figure 6 continued*

lysates and culture media were harvested and probed for CLU immunoreactivity using the anti-Myc antibody. (**C**) pCMV6-mouse CLU Exon 2–9 or Exon 3–9 constructs were transfected into Neuro-2a cells. Cell lysates and culture media were harvested and treated with endoglycosidases. The resulting samples were analyzed for CLU immunoreactivity using the anti-Myc antibody.

DOI: https://doi.org/10.7554/eLife.48255.009

## Human CLU_45 kDa is translated from an AUG (Met) start site in Exon 3

In order to determine whether the expression profiles of human and rodent brain CLU proteins are comparable, human CLU constructs were overexpressed in a human neuroblastoma cell line (SH-SY5Y cells) and analyzed as indicated in the section above. Consistent with all in vivo and in vitro rodent data, the overexpression of human Exon 2–9 results in the expression of mCLU protein isoforms [CLU_60 kDa, CLU_49 kDa, and CLU_37 kDa (*Figure 7A*)]. By contrast, cells overexpressing human Exon 3–9 exhibit a single robust band at 45 kDa (CLU_45 kDa), a finding that is consistent with the presence of an in-frame Met-encoding sequence (AUG) in Exon 3. To further confirm that CLU_45 kDa can be translated from both AUG (Met) and CUG (Leu) start sites, three human Exon 3–9 amino-acid-34 variants were generated: 34_ATG (Met; representing human CLU), 34_CTG (Leu; representing murine CLU), and 34_CTA (Leu; incapable of initiating translation). Constructs containing these variants were transfected into SH-SY5Y cells, and cell lysates were probed for CLU immunoreactivity (*Figure 8A*; lower panel). As expected, a robust band at 45 kDa was expressed in cells transfected with 34_ATG. In addition, though not as robust, a 45-kDa band was also generated following transfection with 34_CTG. However, no 45-kDa band was detected when the translational start site at amino acid 34 was disrupted (34_CTA transfection). These data confirm that CLU_45 kDa is generated in both human and murine cells and that this protein isoform can be generated from both an AUG and a CUG start site in Exon 3.

## Human CLU_45 kDa is not modified by N- or O-linked glycans

Deglycosylation studies were again performed using PNGase F and Endo H to ensure consistency across models and species. As observed in *Figures 6C* and *3C*, CLU_60 kDa is deglycosylated to CLU_49 kDa regardless of the endoglycosidase used, whereas CLU_37 kDa is deglycosylated fully by PNGase F and partially by Endo H (*Figure 7B*; left panel). As Endo H is known to target high mannose glycans, it is worth mentioning that this partial deglycosylation of CLU_37 kDa is inconsistent with deglycosylation data from crude mitochondria. Thus, we suspect that this is probably due to the incomplete maturation of the mCLU protein isoform in our cell model as cells were harvested within 48 hr of transfection. Moreover, consistent with all previous data, the 45 kDa CLU protein isoform generated by Exon 3–9 overexpression was not impacted by treatment with either endoglycosidase. In addition, CLU_45 kDa was not affected by treatment with *O*-Glycosidase (targeting O-linked glycosylation), neuraminidase (targeting sialic acids), or a combination of both enzymes (*Figure 7B*; right panel). Collectively, these data indicate that human CLU_45 kDa is not targeted by N- or O-linked glycosylation.

## Human CLU_45 kDa is localized to the mitochondrial fraction of transfected cells

In order to determine whether the mitochondrial localization of CLU_45 kDa is consistent between human and mouse, cellular fractionation was performed on SH-SY5Y cells overexpressing Exon 3–9. The data indicate low CLU_45 kDa expression in the cytosol, no CLU_45 kDa expression in the PM/ER or the nucleus, and predominant CLU_45 kDa expression in the mitochondrial fraction (*Figure 7C*; left panel). These data confirm the presence of mitochondrial CLU in transfected human cells and demonstrate the consistency between the expression profiles of human and rodent CLU protein.

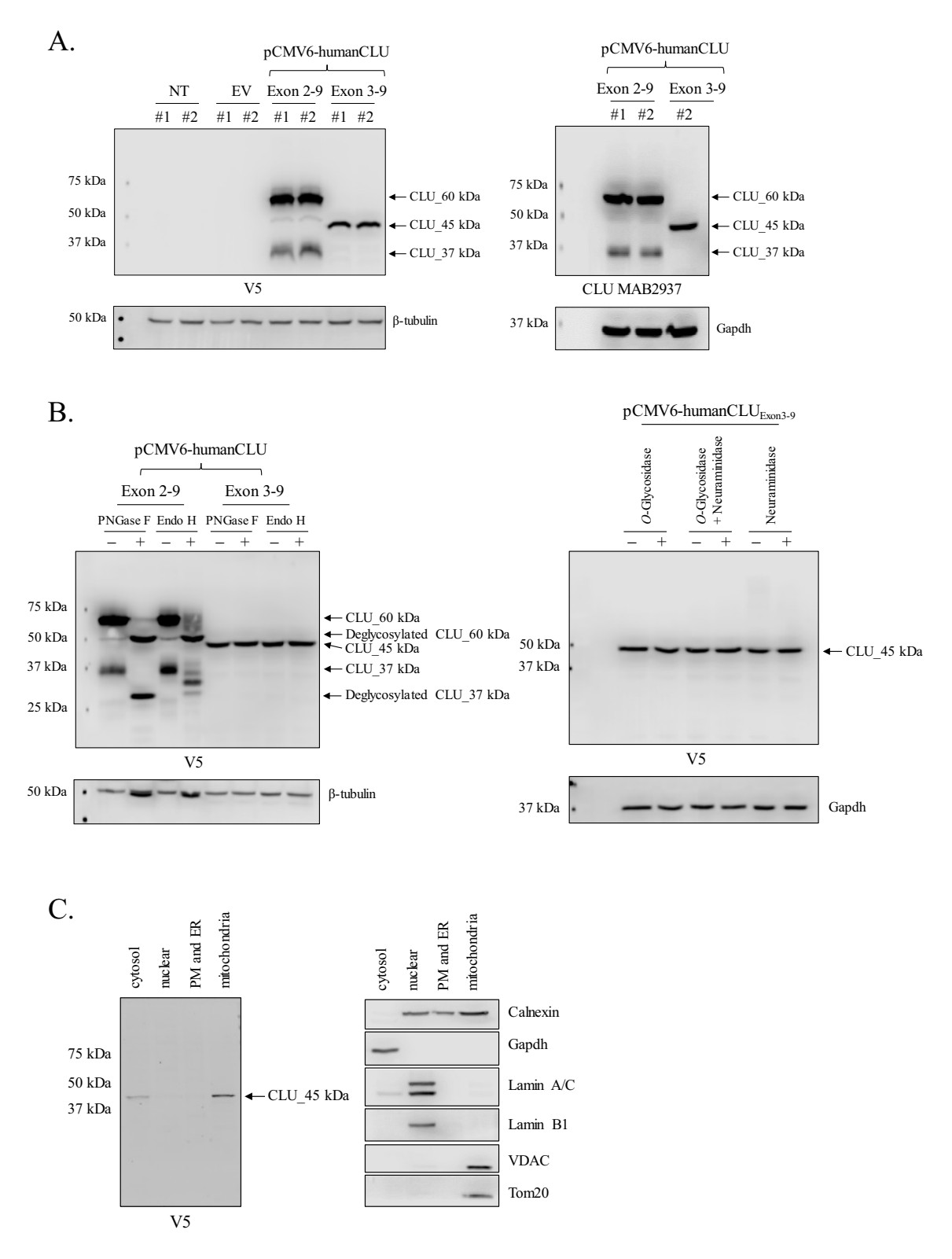

**Figure 7.** Human CLU_45 kDa is translated from an AUG start site in Exon 3. (**A**) Empty vector (pCMV6), V5-tagged pCMV6-human CLU Exon 2–9, or Exon 3–9 constructs were transfected into human SH-SY5Y cells. Cell lysates were harvested and probed for CLU immunoreactivity using anti-V5 (left panel) or anti-Clusterin (right panel). (**B**: left panel) pCMV6-human CLU Exon 2–9 or Exon 3–9 constructs were transfected into human SH-SY5Y cells. The resulting cell lysates were treated with PNGase F or Endo H and probed for CLU immunoreactivity using anti-V5. (**B**: right panel) Cell lysates

*Figure 7 continued on next page*

*Figure 7 continued*

harvested from pCMV6-human CLU Exon 3–9-transfected SH-SY5Y cells were treated with *O*-glycosidase, neuraminidase, or a combination of the two and probed for expression of CLU_45 kDa protein using anti-V5. (**C**) pCMV6-human CLU Exon 3–9-transfected SH-SY5Y cells were subjected to cellular fractionation using the Qiagen Qproteome Mitochondrial Isolation Kit and analyzed for CLU immunoreactivity using anti-V5. Fraction isolation was confirmed using calnexin (ER), Gapdh (cytosol), lamin A/C and lamin B1 (nucleus), and VDAC and Tom20 (mitochondria).

DOI: https://doi.org/10.7554/eLife.48255.010

## Human CLU_45 kDa is expressed in primary human neurons and astrocytes

To determine whether CLU_45 kDa is expressed in human brain cells under physiological conditions, primary human neurons and astrocytes were cultured, and cell lysates were harvested and probed for CLU_45 kDa immunoreactivity (*Figure 8B*). The data indicate the expression of CLU-45 kDa in both cell types. To our knowledge, this is the first publication to demonstrate the expression of CLU_45 kDa in primary human neurons or astrocytes.

## Mitochondrial CLU_45 kDa is expressed in the CLU$^{-/-}$ mouse model

The CLU$^{-/-}$ mouse model was generated in 2001 by *McLaughlin et al. (2000)* as a model for the study of murine autoimmune myocarditis. Since its generation, the CLU$^{-/-}$ mouse model has been utilized to demonstrate the involvement of CLU in several pathological, developmental, and molecular processes (*Imhof et al., 2006*; *Byun et al., 2014*; *Zeng et al., 2015*; *Chayka et al., 2009*; *Ghiggeri et al., 2002*; *Hamada et al., 2011*; *Hong et al., 2016*; *Jiao et al., 2011*; *Kim et al., 2007*; *Mishima et al., 2012*; *Sasaki et al., 2006*; *Savković et al., 2007*; *Seo et al., 2013*; *Shirasawa et al., 2010*), To confirm the absence of CLU in the CLU$^{-/-}$ model, total RNA and whole protein lysate were isolated from age-matched female WT and CLU$^{-/-}$ animals and analyzed for Exon-2- and Exon-3–4-containing mRNA (*Figure 9A*) or CLU immunoreactivity (*Figure 9B*). As expected, Exon 2, which initiates the translation of mCLU protein isoforms (CLU_49 kDa, CLU_60 kDa and the CLU_39 kDa), is absent in the CLU$^{-/-}$ model. However, Exon 3–4 mRNA and the CLU_45 kDa, CLU_49 kDa, and CLU_53 kDa protein isoforms were present, suggesting that the CLU$^{-/-}$ model is not totally devoid of CLU. Although the presence of the 45-kDa protein isoform was expected in these animals on the basis of the location of the targeting cassette used to generate the CLU$^{-/-}$ model, the presence of CLU_49 kDa, which we show to be the Exon-2-initiated non-glycosylated precursor of CLU_60 kDa (*Figure 7B* and *Figure 3C*), was not expected. Moreover, because our initial examination of gene and protein expression in primary neurons and astrocytes suggests that CLU_53 kDa is translated from Exon-1C-containing mRNA, the presence of this isoform was also not predicted. One possible explanation for the presence of CLU_53 kDa would be that this protein isoform is translated from an mRNA transcript containing Exon 1C and Exons 3–9, but not Exon 2, a hypothesis that is consistent with the mRNA and protein data presented here. An exact explanation for the presence of CLU_53 kDa and CLU_49 kDa is not available at this time, but these data demonstrate that the currently available CLU$^{-/-}$ mouse model actually represents a model of mCLU$^{-/-}$. This crucial information will aid in the generation of novel mouse models that may be utilized to examine and compare the effects of mCLU deficiency, mitochondrial CLU deficiency, and total CLU deficiency in the normally aging mouse brain.

## Discussion

Clusterin (CLU) is currently rated as the third most predominant genetic risk factor associated with the development of LOAD (*Alzforum Networking, 2019*). Specifically, a number of clinical studies have confirmed an association between CLU alleles (primarily rs11136000, c allele) and increased risk for LOAD (reviewed in *Woody and Zhao, 2016*), including two of the largest human genome-wide association studies conducted to date (*Harold et al., 2009*; *Lambert et al., 2009*). Although the CLU gene was initially discovered over three decades ago (*Fritz et al., 1983*; *Léger et al., 1987*; *Murphy et al., 1988*), a complete understanding of this gene and the translated protein isoforms is still missing from the current literature. The present study provides a detailed examination of brain CLU in various rodent and human models, including healthy rodent brain tissue, primary rodent and

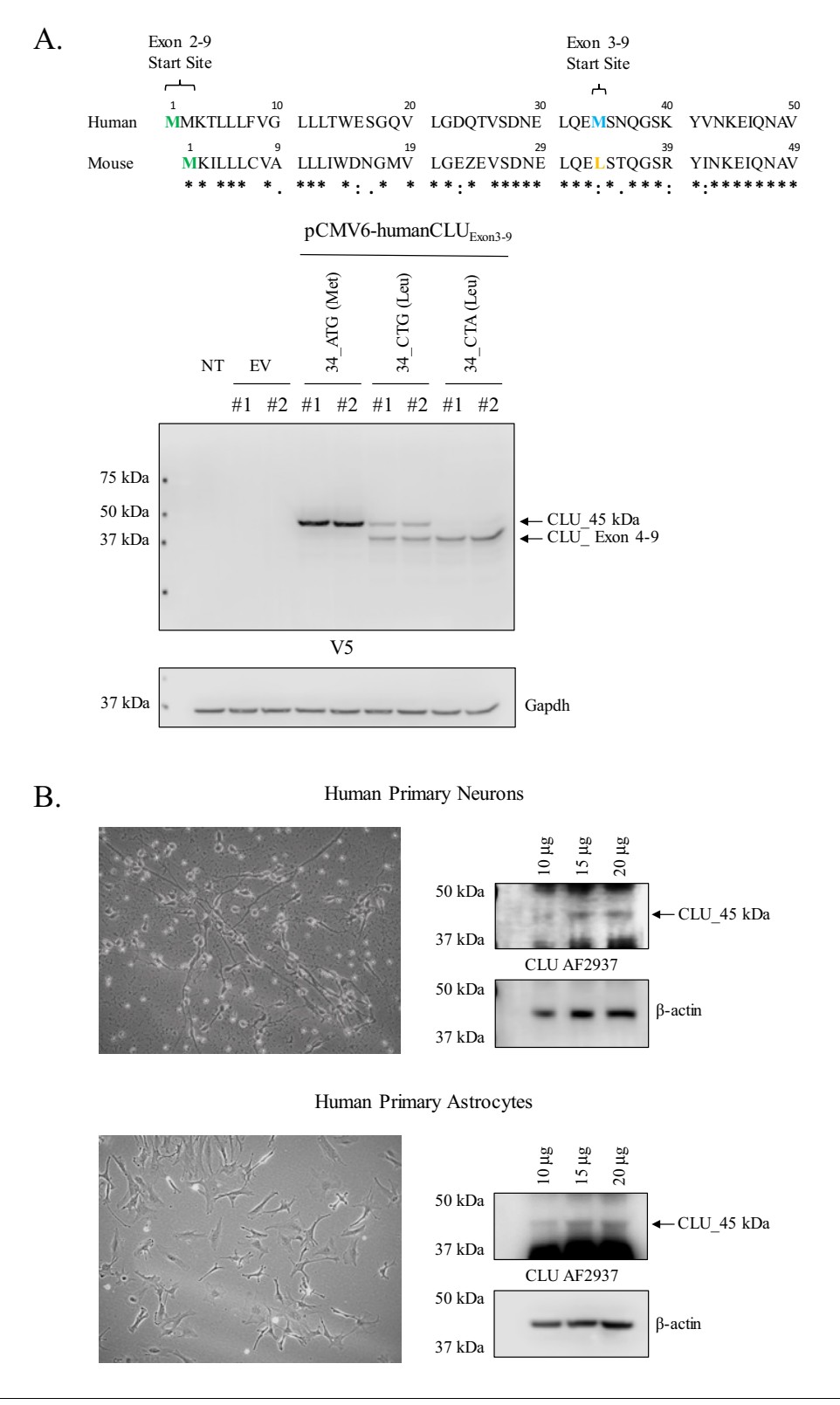

**Figure 8.** Human CLU_45 kDa is translated from an AUG in Exon 3 and is present in human primary neurons and astrocytes. (**A**: upper panel) Alignment of human and mouse CLU: amino-acid sequences of Exons 2 and 3. Green represents the Exon 2 start site in both human and mouse CLU (Met), blue represents the Exon 3 start site for human CLU (Met), and yellow represents the Exon 3 start site for mouse CLU (Leu). (**A**: lower panel) SH-SY5Y cells were transfected with empty vector (pCMV6) or pCMV6-human CLU Exon 3–9 amino-acid 34 variants: 34_ATG (wild-type), 34_CTG (mimics mouse CLU),

*Figure 8 continued*

or 34_CTA (negative control). Cell lysates were probed for CLU immunoreactivity using the anti-V5 antibody. (**B**) Human primary cortical neurons and astrocytes were cultured and imaged to demonstrate cell type visually. Increasing concentrations of whole cell lysate were probed for CLU_45 kDa immunoreactivity using anti-CLU AF2937.

DOI: https://doi.org/10.7554/eLife.48255.011

human neurons and astrocytes, and rodent and human brain-derived cell lines. The data presented herein provide a comprehensive list of CLU protein isoforms and the corresponding mRNA transcripts in the rodent brain, and of the cellular and subcellular localization of each identified isoform. Key among the findings is the identification of a non-glycosylated 45-kDa CLU protein isoform (deemed mitoCLU), which is localized to the mitochondrial matrix of healthy rodent brain tissue and expressed in both rodent and human primary neurons and astrocytes. The data indicate that mitoCLU is translated from Exon 3, specifically from a non-canonical CUG [aa 33 (Leu), murine] or from an in-frame AUG [aa 34 (Met), human], and that mitoCLU is present in the commercially available CLU$^{-/-}$ mouse model. We believe that this foundational knowledge is crucial for our ultimate understanding of the relationship between CLU and the development of LOAD.

## Brain CLU protein isoforms and their localization: clarification of current understandings

The conventional understanding of CLU describes a single secreted, heterodimeric glycoprotein that is upregulated following cellular stress or the onset of neurotoxicity (reviewed in *Valeria Naponelli, 2018*). However, in recent years, alternative CLU protein isoforms have been detected, most predominantly in cancer and cardiovascular research using immortalized human and/or rodent cells. Chief among these protein isoforms is the non-secreted, pro-apoptotic, Exon-2-deficient CLU protein isoform previously referred to as nuclear CLU (nCLU) (*Yang et al., 2000*; *Leskov et al., 2003*) and the briefly researched Exon-5-deficient protein isoform (*Kimura et al., 1997*). Parallel to these findings, we demonstrate that rodent brain expresses multiple CLU immunoreactive bands of various molecular weights (CLU_68 kDa, CLU_60 kDa, CLU_53 kDa, CLU_49 kDa, CLU_45 kDa, and CLU_36 kDa, and CLU_39 kDa) that are located in distinct cellular compartments, findings that provide much needed clarity for multiple aspects of the currently available literature.

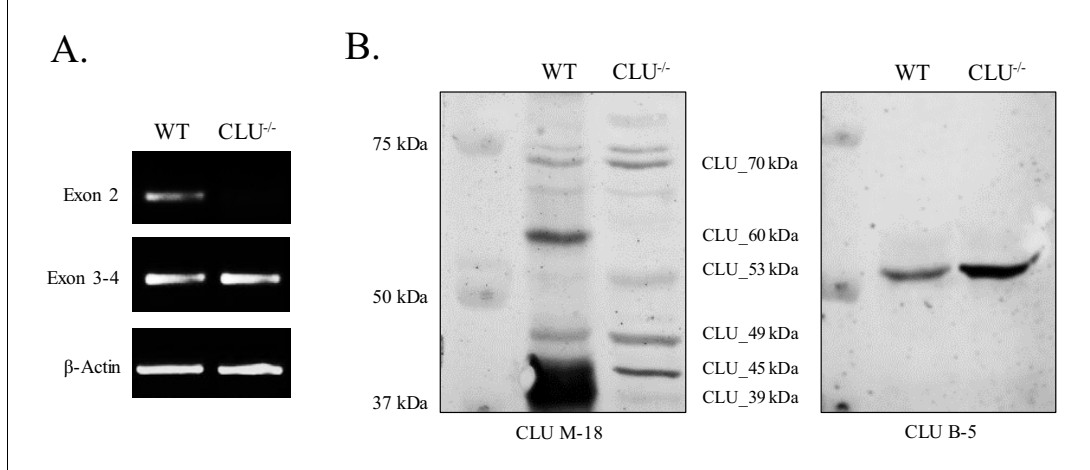

**Figure 9.** Mitochondrial CLU_45 kDa is present in the CLU$^{-/-}$ animal model. (**A**) Total RNA was isolated from WT and CLU$^{-/-}$ cortical tissue. 1.5 µg RNA was reverse transcribed and 50 ng cDNA was analyzed for Exon 2- and Exon 3–4-containing mRNA. To confirm equal loading, β-actin mRNA was amplified using both WT and CLU$^{-/-}$ cDNA. Following amplification, amplicons were run on a 2% agarose gel. (**B**) 30 µg cortical tissue lysate isolated from WT and CLU$^{-/-}$ mice was analyzed by immunoblotting and probed with anti-CLU M-18 (left panel) or anti-CLU B-5 (right panel) overnight at 4°C. Blots were washed and probed with species-specific HRP-conjugate secondary antibodies.

DOI: https://doi.org/10.7554/eLife.48255.012

First, consistent with previous literature (*Yang et al., 2000*; *Caccamo et al., 2003*; *Kim et al., 2012a*; *Reddy et al., 1996*), we observe the expression of one nuclear-localized CLU protein isoform in rodent brain tissue (*Figure 1B*), primary neurons (*Figure 2B*; left panel) and Neuro-2a cells (*Figure 4A*). This finding is further supported by prominent CLU immunoreactivity in the nuclei of neurons located in the hippocampal dentate gyrus (*Figure 1C* and *Figure 1—figure supplement 1B*). However, contrary to previously published data (*Reddy et al., 1996*), the nuclear-localized isoform has a MW of ~68 kDa, not ~45 kDa. Although no 45-kDa protein isoform is present in the nuclear fraction, 45-kDa and 49-kDa protein isoforms are detected in the organelle-containing fraction as well as in the cytosol. These isoforms are designated as mitoCLU and the pre-glycosylated form of mCLU, respectively (discussed later). Thus, our data both confirm and correct current assumptions that are made in the literature pertaining to 'nuclear' CLU. As the 68-kDa nucleus-localized CLU protein isoform was only detected following stringent chemical and mechanical lysis of the nuclear pellet, it is possible that this ~68-kDa protein isoform was not observed in previous studies due to incomplete extraction of nuclear proteins. Moreover, as the 45-kDa and 49-kDa CLU protein isoforms are detected in both the cytosol and the organelles, it is possible that prior detection of the 45-kDa protein isoform in the nuclear fraction was due, in part, to the use of crude cellular fractions for analysis.

Second, our data clarify the cellular localization of brain CLU protein isoforms. Specifically, the data presented here indicate that CLU mRNA and protein are abundant in both neurons and astrocytes (*Figure 2* and *Figure 8*). As relatively few studies have compared CLU mRNA and protein expression profiles between neurons and astrocytes, these data consolidate much of the early research that has provided variable evidence pertaining to brain CLU localization. For example, consistent with previous data (*Charnay et al., 2008*; *O'Bryan et al., 1993*; *Garden et al., 1991*; *Danik et al., 1993*), we observe multiple CLU mRNA transcripts and protein isoforms in the cytosol and the nucleus of primary neurons. In addition, our analysis of astrocytic CLU directly supports previous data that demonstrate clear CLU immunoreactivity in astrocytes and/or other glial cells (*Morgan et al., 1995*; *Imhof et al., 2006*; *Pasinetti et al., 1994*; *Charnay et al., 2012*). The data also identify the CLU mRNA transcripts that are present in neurons and astrocytes [neurons (Exons 1B, 1C, 2, and 3–4); astrocytes (Exons 1A, 1B, 2, 3–4, and low levels of 1C)]. When examined in the context of the neuronal and astrocytic CLU protein expression profile, these data provide key insight into CLU transcription and translation. For example, neurons and astrocytes express CLU_39 kDa, CLU_49 kDa, and CLU_60 kDa as well as mRNA from Exon 1B-3, Exon 2, and Exon 3–4. It has been established that CLU_60 kDa and CLU_39 kDa represent different forms of the mature secreted CLU protein isoform (*Stewart et al., 2007*; *Rohne et al., 2014*) and that mCLU is translated from an AUG (Met) in Exon 2. Analysis of the CLU sequence provided by the NCBI database indicates that this Exon-2-containing mRNA also contains the untranslated region that is known as Exon 1B (*National Center for Biotechnology Information, 2019b* and *Figure 10*). As both cell types contain the mCLU protein isoforms as well as Exons 1B, 2, and 3–4, our mRNA expression data indicate that Exon 1B-9 CLU mRNA translates into CLU_49 kDa (known in the literature as the precursor or pre-protein of mCLU). Furthermore, our deglycosylation data indicate that CLU_49 kDa is glycosylated to form CLU_60 kDa (a high-mannose intermediate glycoprotein) and CLU_36–39 kDa (mCLU subunits modified by complex glycans). In addition, both cell types express CLU_45 kDa. On the basis of the data presented herein, we conclude that this protein isoform is derived from an mRNA transcript beginning in Exon 3 (discussed later). Another noted difference is the presence of Exon 1A in astrocytes but not neurons. The NCBI database indicates that both Exon-1A-containing and Exon-1B-containing mRNA translate into CLU_49 kDa (*Figure 10*). Therefore, our mRNA data indicate that astrocytes contain two mRNA transcripts that are capable of being translated into mCLU, whereas neurons contain only one. These data provide a possible molecular explanation for early publications that postulated that astrocytes were the sole source of secreted mCLU in the brain (*Pasinetti et al., 1994*; *Saura et al., 2003*; *Zwain et al., 1994*). Consistent with this postulation, the astrocytic protein expression profile reveals that astrocytes primarily express mCLU protein isoforms (CLU_49 kDa, CLU_60 kDa, and CLU_36 kDa and/or CLU_39 kDa). Thus, although neurons are capable of producing de novo mCLU, it is highly probable that astrocytes generate and secrete more mCLU than neurons, probably as a means of regulating extracellular homeostasis. This possibility is supported by publications that indicate that astrocyte-secreted mCLU is necessary for maintaining the health of neurons (*Dragunow et al., 1995*; *Cordero-Llana et al., 2011*; *Gregory et al., 2017*).

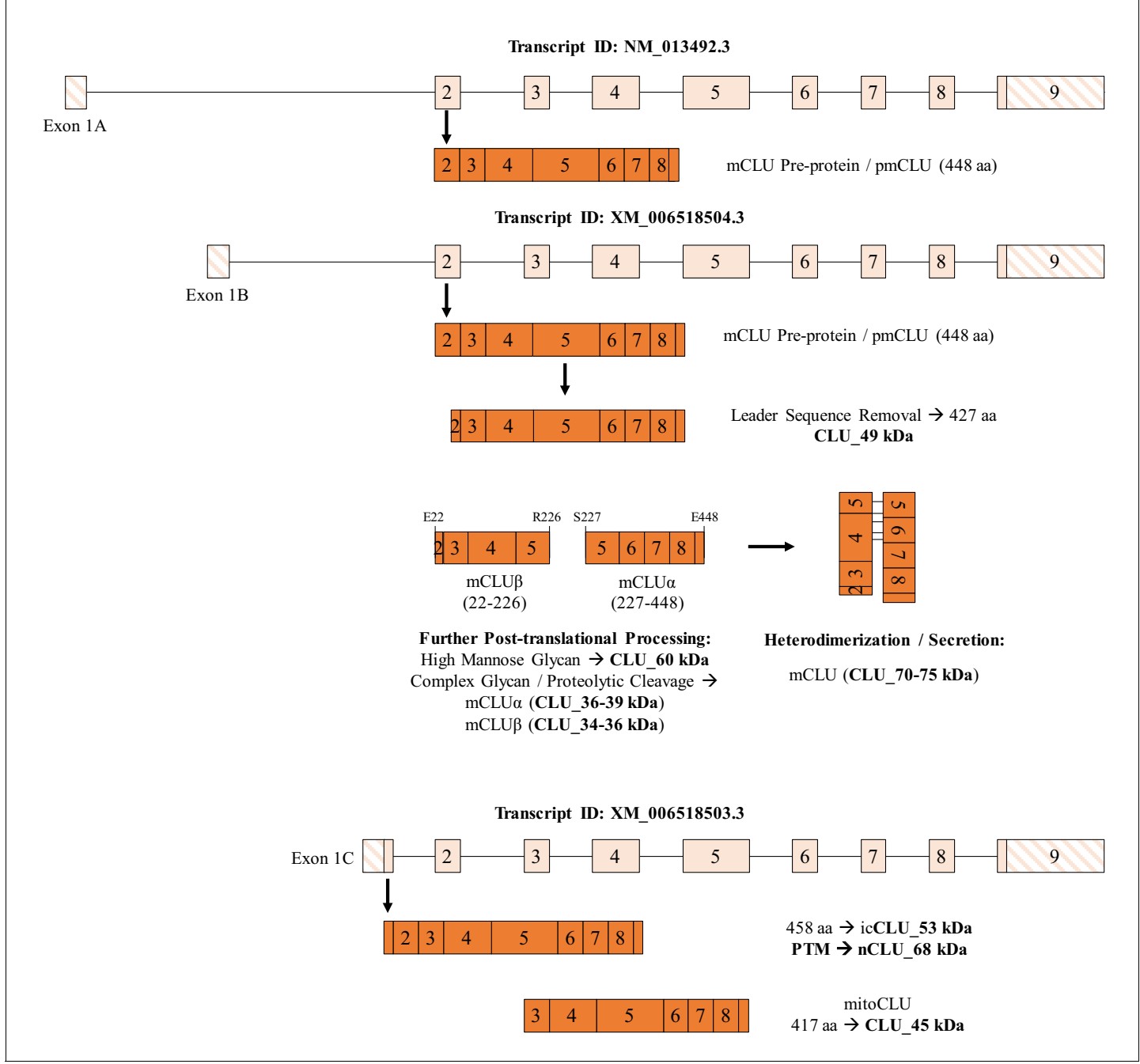

**Figure 10.** Summary of CLU mRNA transcripts and translated protein isoforms in the brain. Three previously described and one newly hypothesized CLU mRNA transcript result in the production of five distinct protein isoforms, including one mature isoform with associated intermediates and three intracellular isoforms. The mCLU pre-protein is generated via translation of Exon-1B-containing mRNA (astrocytes and neurons; middle panel) and Exon-1A-containing mRNA (astrocytes only; upper panel). Once the mCLU pre-protein is translated, the 21-amino-acid ER-targeting sequence (majority of Exon 2) is cleaved and the resultant 427-amino-acid protein (CLU_49 kDa) is glycosylated, first by high mannose glycans (CLU_60 kDa) and then by complex glycans, followed by proteolytic cleavage to form mCLUβ (CLU_34–36 kDa) and mCLUα (CLU_36–39 kDa). Heterodimerization by five disulfide bonds results in the formation of mature CLU (mCLU; CLU_70–75 kDa), which can then be secreted into the extracellular space. In addition, on the basis of our data, we hypothesize that the Exon -C-containing mRNA transcript results in the production of the intracellular CLU protein isoform, icCLU_53 kDa, which is then post-translationally modified to form nCLU_68 kDa. The spliced transcript containing Exons 3–9 results in the production of mitoCLU_45 kDa, which is present in both human and rodent brain mitochondria and is translated from a non-canonical CUG start site in rodents.
DOI: https://doi.org/10.7554/eLife.48255.013

Last, several pieces of data point to the possibility that one or more additional non-secreted CLU protein isoforms are expressed specifically in primary neurons: CLU_53 kDa (*Figure 1B* and *Figure 2B*; left panel) and CLU_68 kDa (*Figures 1* and *2B*; left panel, and 4A). Moreover, CLU mRNA transcript expression indicates more abundant expression of Exon-1C-containing mRNA in primary neurons (*Figure 2A*; right panel). Collectively, these data suggest that CLU_53 kDa and CLU_68 kDa are translated from Exon-1C-containing mRNA in primary neurons, a hypothesis supported by a lack of CLU_53 kDa/CLU_68 kDa expression in cells overexpressing mouse or human pCMV6-CLU Exon 2–9 or pCMV6-CLU Exon 3–9 (*Figures 6–7*). In support of this hypothesis, analysis of the amino-acid sequence generated by the translation of Exon 1 C-9 indicates a length of 458 amino acids with a predicted MW of 52.74 kDa. Therefore, we hypothesize that CLU_53 kDa is translated from an mRNA transcript containing Exons 1 C-9 (*Figure 10*). Furthermore, on the basis of 1) the nuclear localization of CLU_68 kDa, 2) the cytosolic location of CLU_53 kDa, 3) the MW differences between the two isoforms, 4) the importance of ubiquitination and SUMOylation on nuclear protein localization and/or function, and 5) the fact CLU is a known target for ubiquitin, we hypothesize that CLU_53 kDa is post-translationally modified via ubiquitination or SUMOylation to form CLU_68 kDa. However, more research is needed to understand the origin and function of these alternative protein isoforms.

## Mitochondrial CLU is expressed in both human and murine brain and is translated from Exon 3

Preliminary immunostaining of primary cortical neurons indicated overlap between CLU immunoreactivity and the mitochondrial marker Mitotracker (*Figure 3A*), suggesting a possible association between a CLU protein isoform and mitochondria. This finding is congruent with recent literature which demonstrates a mitochondria-associated CLU protein that regulates or interacts with various aspects of the mitochondrial respiratory chain complex-I (MRCC-I) (*Xiao et al., 2019*) and/or the Bax/Bak/Bcl-2 signaling pathway (*Trougakos et al., 2009*; *Zhang et al., 2005*; *Pereira et al., 2018*; *Liu et al., 2018*). The aforementioned publications suggest that the mCLU protein isoform is responsible for these interactions/associations; however, our data indicates that a single CLU protein isoform, CLU_45 kDa (mitoCLU), is expressed in pure rodent brain mitochondria (*Figure 3B*), specifically within the mitochondrial matrix (*Figure 5*). To our knowledge, this is the first publication to demonstrate the expression of a CLU protein isoform in healthy brain mitochondria (rodent or human). The potential impact of this finding as it pertains to the etiology of LOAD is substantial, particularly because mitoCLU was found to be expressed in human primary neurons and astrocytes (*Figure 8B*) and is localized to the mitochondrial matrix — the location of mitochondrial DNA, ribosomes, the tricarboxylic acid cycle, and the enzymes that facilitate mitochondrial respiration. Bioenergetic dysfunction is known to occur throughout the pre-clinical phase of LOAD, with observable reductions in glucose utilization occurring several decades prior to clinical onset (*Mosconi et al., 2009*; *Mosconi et al., 2008*; *Hoyer, 1998*). Therefore, with the discovery of a mitochondria-localized CLU protein isoform, and the knowledge that the CLU AD risk allele is associated with reduced CLU protein levels, several hypotheses can easily be generated. For example, it is plausible that mitoCLU may regulate mitochondrial ribosomes, thus directly affecting mitochondrial protein synthesis and eventually ATP production. Alternatively, mitoCLU may regulate one or more steps in the TCA cycle or may directly influence one or more components of oxidative phosphorylation. However, these hypotheses require further testing before a definitive conclusion can be made.

The discovery of mitoCLU raises several questions pertaining to the mRNA transcript that results in the production of mitoCLU. Our glycosylation studies indicate that although CLU_60 kDa and CLU_36–39 kDa (murine) or 37 kDa (human) are glycosylated, CLU_45 kDa [endogenous (*Figure 3C*) or overexpressed (*Figures 6C* and *7A*)] is not impacted by endoglycosidase treatment, thus indicating a lack of Exon 2 in mitoCLU. On the basis of these data, we hypothesized that mitoCLU was translated from an Exon-3–9-containing CLU mRNA transcript. This hypothesis is consistent with data from human cells, which indicates the presence of an Exon-2-deficient protein isoform translated from amino acid 34 in Exon 3 (*Leskov et al., 2003*; *Reddy et al., 1996*). This hypothesis was confirmed using a series of overexpression constructs [human and murine pCMV6-CLU Exon 2–9 (Exon 2–9) or pCMV6-CLU Exon 3–9 (Exon 3–9)] in both human and murine neuroblastoma cells. Specifically, overexpression of the Exon 2–9 construct in rodent (*Figure 6B*) or human (*Figure 7A*) cells produces all of the previously observed mCLU protein isoforms [CLU_49 kDa, CLU_60 kDa, and

CLU_36–39 kDa (murine) or 37 kDa (human)]. By contrast, CLU_45 kDa was consistently observed in Exon 3–9-transfected cells. Unlike human CLU, murine CLU contains a non-AUG translational start site at amino acid 33 (Leu) (*Starck et al., 2012*), suggesting that mitoCLU is translated from a Leu (CUG) rather than a Met (AUG) start site in rodents, a possibility that is confirmed by the data presented herein (*Figures 6B* and *7A*). As translation is preferentially initiated at an AUG rather than a CUG, this provides a plausible explanation as to why early literature may indicate more robust expression of this previously uncharacterized protein isoform in human cell lines, and/or a complete absence of this isoform in rodent cells and tissues. In rodent models, production of CLU_49 kDa would be predominant due to the type of translational start site present. Therefore, although Exon-3–9-containing mRNA is translated, the Exon 2–9 translated protein product(s) would be present at higher levels in a whole cell or tissue lysate. This supposition was confirmed upon the generation of three human CLU amino acid 34 variants: one representing WT human Exon 3–9 (34_ATG), one representing human CLU with the rodent amino acid 33 Leu codon (34_CTG), and the third containing a Leu codon that is incapable of initiating translation (34_CTA; *Figure 8A*).

## The CLU$^{-/-}$ mouse model represents a model of mature CLU deficiency

Since the generation of the CLU$^{-/-}$ mouse model (*McLaughlin et al., 2000*), this animal model has been utilized to study the impact of CLU deficiency in various pathologies (*Imhof et al., 2006*; *Byun et al., 2014*; *Chayka et al., 2009*; *Hamada et al., 2011*; *Sasaki et al., 2006*; *Seo et al., 2013*; *Shirasawa et al., 2010*; *Zhou et al., 2010*; *Jung et al., 2012*) and developmental processes (*Jiao et al., 2011*; *Kim et al., 2007*). Our lab and others have demonstrated (*McLaughlin et al., 2000*) that the CLU$^{-/-}$ model lacks the mRNA transcript that results in the production of the secreted form of mCLU (CLU_60 kDa and CLU_39 kDa). However, mRNA and protein characterization indicate robust expression of Exon 3–4 mRNA (*Figure 9A*) and mitoCLU protein (Figure 90215624B) in the CLU$^{-/-}$ brain. The presence of mitoCLU was expected on the basis of the location of its translational start site, but the data also indicate the presence of CLU_49 kDa and CLU_53 kDa in CLU$^{-/-}$ brain tissue. It is plausible that CLU_53 kDa may be present in the CLU$^{-/-}$ model, but an explanation for the presence of CLU_49 kDa is currently unavailable. Despite these unexpected findings, the data clearly demonstrate that the previously utilized CLU$^{-/-}$ mouse model is, in actuality, a model of secreted mCLU$^{-/-}$. As several studies utilize this mouse model to understand the overall impact of total CLU deficiency in pathological conditions, these data highlight the importance and a provide rationale for the development of novel mouse models that accurately represent different forms of CLU deficiency.

## Strengths, limitations, and future directions

As the bulk of currently available CLU research is performed in cancer-derived immortalized cell lines and/or models of brain injury or toxicity, a major strength of the present study is the use of multiple physiological models, which provides a more accurate representation of CLU in the brain. Another strength in this study is the multi-antibody strategy that was utilized to generate an accurate representation of the total CLU protein expression profile. In addition, the data presented herein clarify the few inconsistencies between human and mouse CLU mRNA transcription and translation through sequential analysis and comparison of human and murine CLU. However, despite the in-depth examination performed, some deficiencies were noted. First, the subcellular localization of CLU_49 kDa appears to differ between adult cortical tissue and embryonic neurons/astrocytes (*Figures 1B* and *2B*). It is possible that CLU protein isoforms are redistributed as needed through the developmental and aging process to differing subcellular compartments, but more work is needed to confirm this postulation. Second, deglycosylation studies indicate that mCLU protein isoforms are generated via N-linked glycosylation of CLU_49 kDa. However, CLU_49 kDa was detected in animals devoid of Exon 2. Therefore, as no explanation is available, future investigations including an in-depth analysis of CLU expression in the currently available CLU$^{-/-}$ mouse model are needed. Last, the data presented here demonstrate the localization of mitoCLU to the mitochondrial matrix. However, the amino-acid sequence that comprises mitoCLU does not contain a mitochondrial targeting sequence. Therefore, the mechanism of protein recognition and import is currently unknown and should be explored in future studies. In addition, as this is the first publication to identify and characterize mitoCLU conclusively, information pertaining to the function of mitoCLU in the mitochondrial matrix

is not available. As recent studies have indicated that CLU may regulate various aspects of the mitochondrial respiratory chain complex I (*Xiao et al., 2019*), the impact of mitoCLU deficiency on mitochondrial function should be considered in future studies. These proposed studies are expected to further our foundational understanding of CLU physiology and its role in LOAD development.

## Materials and methods

### Animals
The use of animals was approved by the Institutional Animal Care and Use Committee at the University of Kansas and followed NIH guidelines for the care and use of laboratory animals. Tissue/mitochondrial studies were carried out in 4–6-month-old female C57BL/6 (WT) mice. Clusterin-deficient (CLU$^{-/-}$) mice (*McLaughlin et al., 2000*) were derived from three heterozygous breeding pairs purchased from The Jackson Laboratories (JAX stock #005642). For the isolation of primary neurons/astrocytes, pregnant Sprague–Dawley rats were purchased from Harlan Sprague Dawley (Indianapolis, IN). Animals were individually housed under controlled conditions of temperature (18–24°C), humidity (30–70%) and light (12:12 hr light/dark), and received food and water ad libitum.

### Whole tissue protein extraction
30 mg of tissue from indicated brain and peripheral organs was homogenized using the Bullet Blender 24 Homogenizer (Next Advance, Averill Park, NY) with 100 μL 0.5 mm glass beads in variable volumes of tissue protein extraction reagent (TPER, Life Technologies Cat. #78510) supplemented with protease and phosphatase inhibitors according to the manufacturer's instructions.

### Three-buffer tissue fractionation
Cytosolic, organelle, and nuclear proteins were extracted from freshly dissected cortical tissues using a previously described three-buffer extraction system (*Baghirova et al., 2015*) with slight modifications. Briefly, cortical tissues were dissected and minced into smaller sections and homogenized five times with Pestle A in a 2 mL Dounce Homogenizer containing 1 mL Lysis Buffer A [150 mM NaCl, 50 mM HEPES, pH 7.4, 25 μg/mL Digitonin (freshly added)] supplemented with protease and phosphatase inhibitors (PPI). The tissue suspension was then placed in a 1.5 mL centrifuge tube and centrifuged at 500 x g for 10 min at 4°C. The supernatant was placed in a new centrifuge tube and incubated on an end-over-end rocker for 10 min at 4°C. Following incubation, the tissue suspension was centrifuged at 4000 x g for 10 min at 4°C, and the supernatant (cytosolic fraction) was removed and placed in a centrifuge tube. The remaining pellet was re-suspended in 500 μL Lysis Buffer B (150 mM NaCl, 50 mM HEPES pH 7.4, 1% v/v Igepal) supplemented with PPI. Samples were incubated for 30 min at 4°C on an end-over-end rotator followed by centrifugation at 6000 x g for 10 min at 4°C. The supernatant (organelle fraction), which contains proteins from membrane-bound organelles (mitochondria, ER, Golgi, and so on) except those from the nucleus, was stored in a labeled tube at −80°C until use. The remaining pellet was re-suspended in 500 μL Lysis Buffer C (150 mM NaCl, 50 mM HEPES, pH 7.4, 0.5% w/v sodium deoxycholate, 0.1% w/v SDS) supplemented with PPI and incubated for 10 min at 4°C on an end-over-end rocker, followed by centrifugation at 6800 x g for 10 min at 4°C. The supernatant (nuclear fraction) was removed and stored at −80°C until use.

### Primary cortical neurons and astrocytes
Primary cultures of rat cortical neurons were isolated from Day 18 embryonic rat pups as previously described (*Zhao et al., 2004*). Briefly, cortical tissues were dissected from the brains of rat fetuses and treated with 0.02% trypsin in Hank's balanced salt solution (HBSS, Gibco Cat. #14170–112) for 5 min at 37°C. Tissues were dissociated by repeated passage through a series of fire-polished constricted Pasteur pipettes. Cortical neurons were plated onto polyethyleneimine (PEI)-coated 100 mm dishes at a density of $5 \times 10^5$ (biochemical analysis) or onto poly-D-lysine-coated 22 mm glass coverslips (NeuVitro, Cat. #GG-22-PDL) at a density of $2 \times 10^5$ (morphological analysis). Neurons were grown in neuron growth medium [NBM (Gibco Cat. #12348–017), 10 mL B27 supplement (Gibco Cat. #17504–044), 1% Pen-Strep (Gibco Cat. #15140–122), 1.25 mL GlutaMax (Gibco Cat. #35050–061) and 25 μM glutamate] at 37°C in a humidified 5% $CO_2$ atmosphere for the first 3 days and in neuron maintenance medium (neuron growth medium without glutamate) afterwards.

For the isolation of primary cortical astrocytes, euthanasia, dissection, and dissociation were performed as described above for the isolation of primary neurons. Following dissociation, cell suspensions were centrifuged for 10 min at 170 x g and re-suspended in 50 mL astrocyte medium [DMEM/F12 (Gibco Cat. #21041–025), 10% FBS (Gibco Cat. #10437028), 1% Pen-Strep]. Cells were plated at a density of $1 \times 10^6$ in poly-D-lysine-coated T75 flasks and allowed to incubate for 7 days at 37°C in a humidified 5% $CO_2$ atmosphere. On day 7, medium was replaced with fresh astrocyte medium and flasks were incubated for 16 hr on an orbital shaker at a speed of 220 rpm at 37°C. The next day, medium containing dead neurons, oligodendrocytes, and microglia was aspirated, and isolated astrocytes were passaged using 2 mL 0.25% trypsin (Gibco Cat. #25200056) and plated on poly-D-lysine-coated dishes (biochemical analysis) or coverslips (morphological analysis) in astrocyte medium. Astrocytes were harvested on DIV 16 and were passaged two times prior to use.

Primary human neurons (ScienCell Cat. #1520) and human astrocytes (ScienCell Cat. #1800) were purchased from ScienCell Research Laboratories (San Diego, CA). Primary neurons were cultured in neuronal medium (ScienCell Cat. #1521) supplemented with neuronal growth supplement (ScienCell Cat. #1562) and penicillin/streptomycin solution (P/S; ScienCell Cat. #0503). Primary astrocytes were cultured in astrocyte medium (ScienCell Cat. #1801) supplemented with astrocyte growth supplement (ScienCell Cat. #1852), FBS (ScienCell Cat. #0010), and P/S (ScienCell Cat. #0503). Cells were then maintained in a humidified atmosphere containing 5% $CO_2$ at 37°C.

## Subcellular fractionation

Cultured cells were washed 1 time with 1X PBS and were removed from dishes by gently scraping in 1 mL 1X PBS. Cell suspensions were collected and pelleted by centrifugation at 4000 x g for 5 min at 4°C. PBS was aspirated and cell pellets were resuspended in 200 µL cytosolic extraction buffer [CEB, 10 mM HEPES, 60 mM KCl, 1 mM EDTA, and 0.075% (v/v) Igepal]. Samples were incubated for 10 min at 4°C with rotation, passed through a 22 G needle eight times, and centrifuged at 1500 x g for 8 min at 4°C. The supernatant was removed and centrifuged at 8000 x g for 10 min at 4°C, after which the supernatant (cytosolic proteins) was pipetted into a new labeled tube. The remaining pellet from this spin (crude mitochondrial fraction) was resuspended in 30 µL mammalian protein extraction reagent (MPER, Life Technologies Cat. #78503) supplemented with PPI and stored at −80°C. The pellet from the first spin (nuclear proteins) was resuspended in 200 µL nuclear extraction buffer (NEB, 20 mM Tris HCl, 420 mM NaCl, 1.5 mM $MgCl_2$, 0.2 mM EDTA, 25% (v/v) glycerol, 0.5% Igepal) and sonicated for 2 × 10 s with 10 s intervals on ice. Following sonication, the samples were centrifuged at 10,000 x g for 10 min at 4°C. The supernatants were placed in a new labeled centrifuge tube and the pellet was discarded.

## Immunohistochemistry

Free-floating rat hemisphere sections were acclimated to RT. Sections were washed for 2 × 5 min with PBS supplemented with 0.1% Triton X-100 (PBST). Brain sections were incubated in IHC Blocking Buffer [1X PBS supplemented with 5% normal goat serum (NGS, Vector Laboratories Cat. #ZB1027) and 0.3% Triton X-100] at RT for 1 hr. Sections were incubated overnight at 4°C in PBST supplemented with 1% NGS and mouse monoclonal anti-MAP2 (1:1000, Thermo Fisher Cat. #MA1-25043, Lot #RB2160802), rabbit polyclonal anti-Clusterin H-330 (1:500, Santa Cruz Cat. # sc-8354, Lot # F0316), and rat polyclonal anti-GFAP (1:500, Calbiochem Cat. #345860). The following day, brain sections were washed for 3 × 5 min with antibodies containing goat-anti-mouse FITC (1:1000, Abcam Cat. #ab6785, Lot #GR6891-23), goat-anti-rabbit Cy3 (1:1000, Abcam Cat. #ab6939, Lot #GR1836606-3), and goat-anti-rat Cy5 (1:750, Abcam Cat. #ab6565, Lot #GR117398-3). Sections were washed for 3 × 5 min with 1X PBST and mounted on 2' x 3' Subbed Slides immersed in IHC Mounting Medium [2 mL Subbing Solution (0.5 g Gelatin in 100 mL $ddH_2O$, stirred at 60°C until dissolved) + 50 mL 95% EtOH (95 mL 100% EtOH + 5 mL $ddH_2O$) + 6 mL acetate buffer pH 4.9 (15 mL 1 M acetic acid + 35 mL 1 M ammonium acetate) + 6 mL acetate buffer pH 6.0 (1.875 mL 1 M acetic acid + 48.125 mL 1 M ammonium acetate) + 188 mL $ddH_2O$]. Mounted slides were then cover-slipped using Vectasheild mounting medium containing DAPI (Vector Laboratories, Cat. #H-1000) and sealed with clear nail polish. Confocal images were acquired using a customized Olympus IX81/spinning disk confocal inverted microscope (Olympus, Yokogawa) equipped with an Olympus 4X, and 40 × 0.95 NA air objective (Olympus). Images were collected and analyzed using the Slidebook

Software Version 6.0 (Intelligent Imaging Innovations) with 60–80 image stacks with a 0.5 µm step size through the tissue.

## Immunocytochemistry

DIV nine primary cortical neurons or DIV 16 primary cortical astrocytes were incubated in either 250 nM Mitotracker Deep Red FM (Life Technologies, Cat. #M22426, Lot #1654296) or 250 nM Mitotracker Red CMXRos (Life Technologies, Cat. #M7512) for 25 min at 37˚C. Neurons/astrocytes were fixed in freshly prepared 4% paraformaldehyde (PFA, Electron Microscopy Sciences Cat. #15710 s) for 15 min at RT. Cells were washed for 2 × 5 min with RT 1X PBS, permeabilized in 1X PBST for 5 min at RT and blocked in ICC Blocking Solution (1X PBST supplemented with 5% NGS) for 30 min at RT with continual rocking. Cells were stained with the following antibodies in 1X PBST supplemented with 1% NGS as indicated. Neurons: mouse monoclonal anti-MAP2 (1:750) and rabbit polyclonal anti-Clusterin H-330 (1:500). Astrocytes: rat polyclonal anti-GFAP (1:500) and rabbit polyclonal anti-Clusterin H-330 (1:500). The next morning, samples were washed for 3 × 5 min with 1X PBS and incubated for 1 hr at RT in 1X PBST supplemented with 1% NGS and the fluorophore conjugated secondary antibodies Goat anti-rabbit Cy3 (Clusterin; 1:1000) and Goat anti-mouse FITC (MAP2; 1:1000) or Goat anti-rat Cy5 (GFAP; 1:1000). Following immunolabeling, all samples were washed for 3 × 5 min with 1X PBS and mounted on glass microscopy slides using Vectashield Hardset mounting medium containing DAPI (Vector Laboratories Cat. #H-1200). Confocal images were acquired using a customized Olympus IX81/spinning disk confocal inverted microscope equipped with an Olympus 40 × 0.95 NA air objective or an Olympus 60X NA 1.42 oil objective (Olympus). Images were collected and analyzed using the Slidebook Software Version 6.0 with 15–20 image stacks with a 0.1 µm step size through the cells.

## RNA isolation

Total RNA was isolated from primary neurons/astrocytes/tissues using the commercially available Pure Link RNA Mini Kit according to the manufacturer's instructions (tissues) or with slight modifications (cells). Briefly, 500 µL of TRIzol Reagent (Life Technologies, Cat. #15596026) was added to each culture dish of primary neurons/astrocytes and cells were scraped and transferred to a 1.5 mL centrifuge tube at RT for 5 min. 200 µL chloroform (MP Biomedicals, Cat. #2194002) was added to each tube and tubes were mixed by inversion for 15 s. Samples were centrifuged at 12,000 x g for 15 min at 4˚C, and ~400 µL of the upper aqueous portion was pipetted into a new microcentrifuge tube. 1 vol of 70% ethanol was added to each sample and mixed by vortexing at max speed for 15 s. RNA was then purified using the PureLink RNA Mini kit according to the manufacturer's instructions.

## Reverse transcription and RT-qPCR

1 µg total RNA was reverse transcribed using the Applied Biosystems High Capacity RNA-to-cDNA Kit (Applied BioSciences, Cat. # 4387406) according to the manufacturer's instructions. Exon 1A-3, Exon 1B-3, Exon 1 C-3, Exon 2, and Exons 3–4 (total CLU) gene expression was evaluated in 10 µL reactions containing 5 µL PowerUP SYBR Green MasterMix (Life Technologies, Cat. # A25741), 25 ng cDNA, 500 nM of forward and reverse primers, and ultrapure ddH2O. In all gene expression studies, at least two control genes were also amplified and the gene that remained more consistent across groups was utilized: in this case Gapdh (neurons/astrocytes) and β-Actin (tissues). Cycling conditions used are as follows: 50˚C for 2 min, 95˚C for 10 min followed by 40 cycles of 95˚C for 15 s and 65˚C for 1 min. Using data provided by the NCBI database (*National Center for Biotechnology Information, 2019c*), PCR primers were designed to amplify Exon 1A, Exon 1B, Exon 1C and Exon 2 specifically (*Figure 2A*). In addition, as all CLU mRNA isoforms are sequentially homologous from Exons 3–9, primers were developed to amplify total CLU by targeting a region in Exon 3–4 (*Figure 2A*). Following amplification, PCR products were visualized using a 2% agarose gel. Primers used in the studies presented herein are as follows: Exon 1A-3: Forward: 5'- CACCTCTAGGC TTCCAGAAA-3', Reverse: 5'-CTCTTTCTTCTTCTTGGCTTCC-3'; Exon 1B-3: Forward: 5'-AGTCCATA TTACCAGCAGACC-3', Reverse: 5'-AGTTTTTATGTGCTTCACTCCC-3'; Exon 1 C-3: Forward: 5'-TCC TAAATTCCTTCCCTTCCC-3', Reverse: 5'-AGTTTTTATGTGCTTCACTCCC-3'; Exon 2: Forward: 5'-GCTGCTGATCTGGGACAATG-3', Reverse: 5'-ACCTACTCCCTTGAGTGGACA-3'; Exon 3–4:

Forward: 5′-CCTTGCTCAACAGTTTAGAGGAA-3′, Reverse: 5′-CATCATGGTCTCGGTACACACTT-3′; Gapdh: Forward: 5′-AGGTCGGTGTGAACGGATTTG-3′, Reverse: 5′-TGTAGACCATGTAGTTGAGG TCA-3′; β-Actin: Forward: 5′-GTGACGTTGACATCCGTAAAGA-3′′, Reverse: 5′-GCCGGACTCA TCGGACTCC-3′.

## Isolation of pure mitochondria

4–6-month-old female WT mice were euthanized (two mice/isolation) and whole cortical tissues were dissected. Brain sections were washed for 10 s in 20 mL ice cold mitochondria isolation buffer (MIB, 320 mM sucrose, 1 mM EDTA, 10 mM tris-HCl, pH adjusted to 7.4) without PPI or BSA. Tissues were then minced into 8–10 small pieces, transferred to a 5 mL glass dounce homogenizer containing 5 mL ice cold MIB supplemented with PPI and 0.05% BSA (added fresh), and gently homogenized using 10 slow strokes on ice. The homogenate was transferred to a 30 mL polycarbonate tube on ice and centrifuged at 1330 x g for 8 min at 4°C (Spin 1). The supernatant was pipetted into a separate polycarbonate tube and placed on ice. Homogenization and centrifugation were repeated on the pellet (Spin 2). Supernatants from Spin 1 and Spin 2 were combined and centrifuged once more at 1330 x g for 8 min at 4°C (Spin 3). A 100 µL aliquot was removed from the supernatant and the remaining supernatant was centrifuged at 21,200 x g for 10 min at 4°C (Spin 4). The supernatant was discarded and the remaining pellet was resuspended by pipetting in 6 mL 15% percoll. The resuspended pellet was centrifuged at 21,200 x g for 10 min at 4°C (Spin 5). Using a P-1000 pipette, the loose pellet of 'crude' mitochondria (~1–2 ml) on the bottom of the tube was layered onto a premade percoll 23%/40% discontinuous percoll gradient. The gradient was centrifuged at 40,000 x g for 18 min at 4°C (Spin 6). The entire cloudy white interface between the two percoll layers (up to the upper layer of percoll) was removed with a glass Pasteur pipette and transferred into a prechilled 15 mL polycarbonate tube with ice cold MIB (without PPI). Mitochondrial samples were centrifuged at 21,700 x g for 13 min at 4°C (Spin 7). The loose pellet at the bottom of the tube was removed and pipetted into a sterile labeled 1.5 mL centrifuge tube and centrifuged at 6600 x g for 8 min at 4°C (Spin 8) in a table top centrifuge. The supernatant was removed and the remaining mitochondrial pellets were re-suspended in 50–100 µL MIB. Protein concentration was determined via BCA assay and samples were analyzed as indicated.

## Digitonin mitochondrial sub-fractionation

Pure mitochondria were re-suspended in 200 uL Mitochondria sub-fractionation buffer (MSF Buffer, 220 mM mannitol, 70 mM sucrose, 2 mM HEPES, pH adjusted to 7.4 with KOH) supplemented with 0.5 mg/mL BSA. The mitochondrial suspension was then separated into four 50 µL aliquots and 50 µL of MSF buffer containing 0%, 0.2%, 0.4%, or 0.6% digitonin (diluted from a 1.6% stock solution in MSF buffer) was added to respective tubes to yield a final concentration range of 0–0.3% digitonin (100 µL total volume). Tubes were incubated for exactly 5 min on ice and quickly diluted with 100 µL cold MSF buffer. Samples were transferred to polypropylene tubes and centrifuged at 50,000 x g for 25 min at 4°C in a Beckman SW-41 Ti Swinging Bucket Rotor. Supernatants were carefully removed and the pellet was resuspended in 30 µL MPER supplemented with PPI.

## Cell lines

Mouse Neuro-2a (AATC CCL-131) and human SH-SY5Y cells (AATC CRL-2266) were purchased from ATCC (Manassas, VA) and authenticated using STR profiling by IDEXX BioAnalytics (Columbia, MO). The authentication certificate is provided as *Supplementary file 1*. In addition, both cell lines were tested for mycoplasma contamination using the PCR-based Venor GeM Mycoplasma Detection Kit (Sigma, Cat. #MP0025) following the manufacturer's instructions. Negative results indicated that no mycoplasma contamination was present in the cell lines used in the studies presented herein. Representative testing results from two different passages are provided as *Supplementary file 2*.

## Neuro-2a cells and siRNA-mediated knockdown

Neuro-2a cells were cultured according to the established guidelines. Briefly, cells were maintained in glucose containing (4.5 g/L D-glucose), sodium pyruvate-free Neuro-2a maintenance medium (DMEM, 10% FBS, 1% Pen-Strep). siRNA constructs targeting rodent CLU (Exon 3, Life Technologies, Cat. #4390771, siRNA Construct #s201172) and non-targeting scramble siRNA (negative

control, Life Technologies, Cat. #AM4611) were transfected at a concentration of 200 pM into Neuro-2a cells [$1 \times 10^5$ cells/mL (2 mL/well)] using Lipofectamine RNAiMax transfection reagent (Life Technologies, Cat. #13778100) according to the manufacturer's instructions. 72 hr post-transfection, cells were harvested and analyzed for CLU immunoreactivity via SDS-PAGE. Sequences for the CLU targeting siRNA constructs are as follows: Exon 3 targeting — sense: 5′-GCUCAACAGUUUAGAG-GAAtt-3′; antisense: 5′-UUCCUCUAAACUGUUGAGCaa-3′.

## Enzymatic deglycosylation

Deglycosylation of CLU protein isoforms were achieved using PNGase F (New England BioLabs or NEB Cat. #P0704), Endo H (NEB Cat. #P0702), O-glycosidase (NEB Cat. #P0733), and α2–3,6,8 neuraminidase (NEB Cat. #P0720). 20 μg of total protein sample was first denatured by incubation with glycoprotein denaturing buffer for 10 min at 95℃. Denatured protein sample was then treated with glycosidase enzyme in GlycoBuffer, as specified by the manufacturer, for 3 hr at 37℃. For α2–3,6,8 neuraminidase treatment, 20 μg of total protein sample was incubated with 100 units of the enzyme in a final volume of 20 μL for 16 hr at 37℃. The reactions were stopped by mixing with Laemmli sample buffer containing 2-ME and analyzed by Western blot as indicated below. RNase B (NEB Cat. #P7817) and fetuin (NEB Cat. #P6042) were used as positive controls for the enzymes and the data are located in Supplementary Figure D.

## Generation of overexpression constructs

pCMV6-mouse CLU Exon 2–9 (C-terminal Myc-DDK-tagged) was purchased from OriGene (Cat. #MR207148). The mouse CLU Exon 3–9 was amplified by PCR from cDNA of the mouse CLU Exon 2–9 plasmid using two oligonucleotide primers, mouse CLU Exon 3–9 forward [5′-GACGGCGATCGCCGAACTGTCCACTCAAGGGAGT-3′ (AsiSI restriction site is underlined)] and mouse CLU Exon 3–9 reverse [5′-GGTCCTCGAGTTCCGCACGGCTTTTCCT-3′ (XhoI restriction site is underlined)]. AsiSI and XhoI sites were incorporated into the forward and reverse primer for cloning into the pCMV6-Entry expression vector (OriGene, Cat. #PS100001). Human CLU Exon 2–9 (with C-terminal V5 tag) cloned into the pLX304 vector was purchased from DNASU Plasmid Repository (Cat. #HsCD00435174). The human CLU-V5-tagged DNA was then lifted out by PCR and cloned into pCMV6-Entry vector at SgfI and XhoI sites. The human CLU Exon 3–9 was amplified by PCR from cDNA of the human CLU Exon 2–9 plasmid using two oligonucleotide primers, human CLU Exon 3–9 forward [5′-GACGGCGATCGCCGAAATGTCCAATCAGGGAAGT-3′ (SgfI restriction site is underlined)] and human CLU Exon 3–9 reverse [5′-GGTCCTCGAGTTACTACGTAGAATCGAG ACCGAG-3′ (XhoI restriction site is underlined)]. The methionine residue at 34 in human CLU (translation start residue of Exon 3–9) was mutated to leucine by PCR to produce aug34cug (Leu initiation codon) or aug34cua (Leu non-initiation codon), using the following primers: aug34cug forward [5′-GACG GCGATCGCCGAA**CTG**TCCAATCAGGGAAGT-3′ (the AsiSI restriction site is underlined and the Leu (initiation codon) is bolded)], aug34cua forward [5′-GACGGCGATCGCCGAA**CTA**TCCAATCAGGGA AGT-3′ (the SgfI restriction site is underlined and the Leu (non-initiation codon) is bolded)], and human CLU Exon 3–9 reverse. The SgfI-XhoI fragment from the PCR amplicon of human CLU Exon 3–9 was sub-cloned into the same site of pCMV6-Entry. All sequences were verified by DNA sequencing by GENEWIZ (South Plainfield, NJ).

## Plasmid amplification, purification, and transfection

Following confirmation of successful plasmid generation, plasmids were transformed into 5-alpha competent E. coli (NEB, Cat. #C2987) in accordance with the manufacturer's instructions. Transformed bacteria were plated in agarose plates containing the appropriate vector-specific antibiotic and incubated for up to 16 hr at 37℃. Colonies were picked and placed in LB Broth containing the appropriate antibiotic. Inoculated broth was incubated at 37℃ with shaking overnight. Plasmid DNA was purified using the QIAGEN MIDI Prep Kit (Cat. #12643) according to the manufacturer's instructions. Following confirmation of plasmid isolation, cells were transfected with the indicated concentrations of plasmid using jetPRIME (Polyplus Transfection, Cat. #114–07) according to the manufacturer's instructions.

## Immunoblotting

20–30 µg of total protein sample isolated from tissue, cells or mitochondrial samples was resolved via reducing SDS-PAGE. Gels were run for 15 min at 125 V followed by 40 min at 180 V in 1X Western Blot running buffer [100 mL 10X Western Blot running buffer (250 mM Tris Base, 1.92 M Glycine) + 890 mL ddH$_2$O + 10 mL 10% SDS (10 g SDS powder + 90 mL ddH$_2$O)]. Resolved proteins were then transferred to 0.2 µm pore-sized PVDF membranes for 1 hr on ice in 1X Western Blot transfer buffer (100 mL 10X Western Blot running buffer + 200 mL 100% methanol + 700 mL ddH$_2$O) and blocked with 2% non-fat milk in 1X TBST [100 mL 10X TBS (200 mM Tris, 1.5 mM NaCl, pH 7.6) + 890 mL ddH$_2$O + 10 mL 10% Tween-20 solution] for 1 hr at RT. Membranes were incubated with customized dilutions of the indicated primary antibodies overnight at 4°C followed by incubation with the appropriate HRP-conjugated secondary antibodies for 1 hr at RT. Membranes were washed with 1X TBST for 3 × 10 min at RT with continual rocking, and then visualized using chemiluminescence with the Clarity Western ECL substrate (BioRad, Cat. #1705061) and scanned using the C-DiGit blot scanner (LI-COR Biosciences). Primary antibodies used in the studies presented herein include: rabbit polyclonal actin beta (β-actin, Pierce, Cat. #PA5-16914), mouse monoclonal beta-tubulin [TBN06 (Tub 2.5)] (β-tubulin, Pierce, Cat. #MA5-11732), rabbit polyclonal calnexin (H70) (Santa Cruz, Cat. #sc-11397), rabbit monoclonal calreticulin (D3E6) (Cell Signaling, Cat. #12238), rabbit polyclonal clusterin (CLU H-330; Santa Cruz, Cat. #sc-8354), goat polyclonal clusterin-α (CLU M18; Santa Cruz, Cat. #sc-6420), mouse monoclonal clusterin-α (CLU B-5; Santa Cruz, Cat. #sc-5289), mouse monoclonal human clusterin antibody (CLU MAB2937; R and D Systems, Cat. #MAB2937), goat polyclonal human clusterin isoform one antibody (CLU AF2937; R and D Systems, Cat. #AF2937), rabbit monoclonal COX IV (3E11) (Cell Signaling, Cat. #4850), rabbit monoclonal cytochrome C (Cyt. C; Cell Signaling, Cat. #119402), mouse monoclonal Gapdh (Santa Cruz, Cat. #sc-3223), mouse monoclonal Grp75 (BioRad, Cat. #VMA00084T), rabbit monoclonal Hsp60 (Cell Signaling, Cat. #12165), mouse monoclonal lamin A/C (Cell Signaling, Cat. #4777), mouse monoclonal lamin B1 (Santa Cruz, Cat. #sc-365962), rabbit polyclonal MYC-tag (OrigGene, Cat. #TA150081), mouse monoclonal Tom20 (Santa Cruz, Cat. #sc-136211), rabbit monoclonal V5-tag (Cell Signaling, Cat. #13202S), and rabbit monoclonal VDAC (Cell Signaling, Cat. #4661). The secondary antibodies that were used include: goat anti-mouse IgG (H+L) HRP (Pierce, Cat. #31430), goat anti-rabbit IgG (H+L) HRP (Pierce, Cat. #31462), and rabbit anti-goat IgG (H+L) HRP (Pierce, Cat. #31402).

For better visualization of protein isoform molecular weights, membranes were stained using the 3,3′,5,5′-tetramethylbenzidine (TMB) peroxidase (HRP) substrate kit (Vector Laboratories, Cat. #SK-4400). Following ECL acquisition, membranes were washed with 1X TBST for 10 min at RT with constant rocking. While the membrane was being washed, TMB solution was generated by diluting 2 drops of buffering solution, 3 drops of TMB substrate, 2 drops of stabilizing solution, and 2 drops of H$_2$O$_2$ in 5 mL ddH$_2$O. TBST was dumped and the residual buffer was removed with a P1000 pipette. TMB solution was added and the membrane was incubated at RT with continual rocking. Following incubation, TMB solution was removed and the membrane was washed, dried, and scanned using a standard scanner.

## Acknowledgements

This research was supported by grants from the National Institutes of Health (NIH; R21AG055964, R21AG059177, R01AG061038), the NIH-funded KU Alzheimer's Disease Center (P30AG035982), the NIH-funded K-INBRE Program (P20GM103418), and a pre-doctoral fellowship from the American Foundation of Pharmaceutical Education (AFPE; to SKH).

## Additional information

### Funding

| Funder | Grant reference number | Author |
| --- | --- | --- |
| National Institutes of Health | R01AG061038 | Liqin Zhao |
| National Institutes of Health | R21AG055964 | Liqin Zhao |
| National Institutes of Health | R21AG059177 | Liqin Zhao |

| American Foundation for Pharmaceutical Education | Pre-doctoral Fellowship | Sarah K Herring |
| National Institutes of Health | P30AG035982 Pilot Project | Liqin Zhao |
| National Institutes of Health | P20GM103418 Pilot Project | Liqin Zhao |

The funders had no role in study design, data collection and interpretation, or the decision to submit the work for publication.

### Author contributions

Sarah K Herring, Performed and interpreted most of the experiments, Wrote the initial draft of the manuscript; Hee-Jung Moon, Performed the overexpression studies including construct generation, Assisted the preparation of the manuscript; Punam Rawal, Performed the analysis of CLU protein isoform expression in human primary neurons and astrocytes; Anindit Chhibber, Assisted with experimental methodology and antibody validation; Liqin Zhao, Conceived and supervised the project, Edited and finalized the manuscript

### Author ORCIDs

Liqin Zhao https://orcid.org/0000-0002-0491-6943

### Ethics

Animal experimentation: The use of animals was approved by the Institutional Animal Care and Use Committee (IACUC) at the University of Kansas (Animal Use Statement # 220-04) and followed NIH guidelines for the care and use of laboratory animals.

### Decision letter and Author response

Decision letter https://doi.org/10.7554/eLife.48255.016
Author response https://doi.org/10.7554/eLife.48255.017

---

# Additional files

### Supplementary files

• Supplementary file 1. Cell line authentication certificate.
DOI:
• Supplementary file 2. Mycoplasma testing of cell lines at different passages.
DOI:
• Transparent reporting form DOI: https://doi.org/10.7554/eLife.48255.014

### Data availability

All data generated or analyzed during this study are included in the manuscript.

---

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
