## [Decision Letter]

**Acceptance summary:**

This work provides a detailed characterization of mitochondria-targeted brain clusterin protein isoform in neuronal and glial cells. It provides a foundation for the future studies on its role in neurophysiology on one side, and in the pathogenesis of Alzheimer's disease, on the other side.

**Decision letter after peer review:**

Thank you for submitting your article "Brain Clusterin (ApoJ) Protein Isoforms and Mitochondrial Localization" for consideration by *eLife*. Your article has been reviewed by two peer reviewers, and the evaluation has been overseen by a Reviewing Editor and Huda Zoghbi as the Senior Editor. The reviewers have opted to remain anonymous.

The reviewers have discussed the reviews with one another and the Reviewing Editor has drafted this decision to help you prepare a revised submission.

Summary:

The authors have performed a thorough characterization of CLU protein, which is the third most prominent genetic risk factor in the development of late-onset Alzheimer's disease. By the use of immunoblots, they confirmed the existence of several CLU isoforms, which, up to now, have been poorly characterized. A 45 kDa isoform has been demonstrated to exist in humans, but not in mice. The authors convincingly demonstrated that CLU_45 kDa protein isoform is also present in C57BL/6 mouse brains. They assumed that the CLU_45 kDa protein is present in lower levels in mice, presumably because the putative translation start site is a non-AUG codon, although they do not provide hard evidence for these claims. They proved that the CLU_45 kDa protein does not seem to have any large glycosylation patterns and they showed that it localizes to the mitochondrial matrix in female C57BL/6 mouse brains. Furthermore, they demonstrated an important flaw in commercially available CLU^-/-^ mouse models as some CLU isoforms are still present in this model (CLU_45 kDa protein isoform, CLU_53 kDa and CLU_49 kDa).

Overall, this manuscript has the potential to significantly add to current knowledge of CLU, and the experiments seem to be rigorously conducted. Nonetheless, we do have some comments to improve the manuscript, which you can find below.

Essential revisions:

- The manuscript is too long, and needs to be shortened. Many methods are described in the Results section reducing the flow of the manuscript. Also, the titles of the Results sections are vague, e.g. "exhibit distinct subcellular location". This should be more concise yet more specific. In general, the sentences should be shorter; currently, they are too complicated for a busy reader to quickly grasp the essential concepts. All sections, including the Introduction, Results, and Discussion, should be shortened. Similarly, figure legends include methods which are unnecessary and misleading as the reader cannot quickly access the information regarding the results.

- It is important that the (raw) data that was not shown in the manuscript would be available (preferably annotated) somewhere in publicly available repositories and/or supplementary material; e.g. "it possible that this isoform is detected due to the presence of Exon 1C mRNA in the knockout model (data not shown)."

- "Of particular significance, we discovered for the first time a novel CLU isoform that was localized to the mitochondrial matrix and was translated from Exons 3-9." This sentence in the Abstract is misleading, because after reading the article, it becomes apparent that the 45 kDa isoform has already been detected in human cell lines before, but was never characterized.

- Figure 1A: the authors claimed the lack of differences in the CLU protein expression profiles between male and females. However, the variance in some bands is striking, specifically in lungs and heart. It would be desirable to know if the authors analyzed these samples in the same gel in order to justify their conclusion. Moreover, as the authors remarked, LOAD more frequently impacts the female population, hence, it could be interesting to decipher whether the cellular localization of the different CLU isoforms vary between female and male brains, given that both sexes have similar protein expression profile in the brain. As claimed by published study from this group, clusterin could be a link between hypometabolic phenotype and increased amyloid dyshomeostasis in early onset of AD in female mice (Zhao et al., Neurobiology of aging, 2016).

- Figure 1C. It is unclear which brain region (and consequently cell type) is analyzed in this immunohistochemistry experiment. Moreover, since the authors investigate the cell type-specific location of brain CLU proteins, I consider that a whole brain section should be tested with the same technique for this aim.

- The deglycosylation studies performed by the authors to claim that the proposed mitochondrial isoform of CLU is not glycosylated, consist in differences of molecular weight in protein immunoblots. However, some glycosylation patterns can be small, e.g. many O-linked glycosylation, not necessarily clear in a gel. Thus, I suggest to conduct a more robust experiment for post-translational modifications.

- The authors declared that human CLU_45 kDa is translated from an AUG start site in Exon 3. This conclusion is solely based on overexpression of an Exon 3-9 construct in human SH-SY5Y cells. Here, I would suggest to refer to previously published evidence that this protein isoform is indeed expressed at basal levels in human cell lines and to show more evidence of this AUG codon being, indeed, the translation start site. There is the possibility that this protein isoform is also translated from a non-canonical start site, or that the translation is even initiated from another exon. The authors should perform a transfection experiment with a mutated AUG site to show that the AUG codon on Exon 3 is effectively the translation start site.

- "Moreover, as CLU_45 kDa is generated from a canonical translational start site in humans and is therefore likely translated at higher levels in the human brain, these data highlight the potential importance of mitochondrial CLU in human brain." This claim is overstating due to the lack of hard evidence about the nature of the translational start site and about the supposed higher level of translation in the human brain.

- "Moreover, we have identified a novel 45 kDa mitochondrial CLU protein isoform (deemed mitoCLU) present in healthy rodent brain tissue and both human and rodent brain-derived cells." "To our knowledge, this is the first publication to conclusively demonstrate the expression of a CLU protein isoform in healthy brain mitochondria (rodent or human)." The authors claim the expression of the mitoCLU isoform in human brains, while they only detected it in an overexpression study in human cell cultures.

---

## [Author Response]

Essential revisions:- The manuscript is too long, and needs to be shortened. Many methods are described in the Results section reducing the flow of the manuscript. Also, the titles of the Results sections are vague, e.g. "exhibit distinct subcellular location". This should be more concise yet more specific. In general, the sentences should be shorter; currently, they are too complicated for a busy reader to quickly grasp the essential concepts. All sections, including the Introduction, Results, and Discussion, should be shortened. Similarly, figure legends include methods which are unnecessary and misleading as the reader cannot quickly access the information regarding the results.

The manuscript has been substantially revised to a reasonable length. Several of the sections in the manuscript have been consolidated to improve the flow. Sentences have been shortened and section titles have been altered. Figure legends have also been updated to allow the reader to quickly grasp the data conveyed.

- It is important that the (raw) data that was not shown in the manuscript would be available (preferably annotated) somewhere in publicly available repositories and/or supplementary material; e.g. "it possible that this isoform is detected due to the presence of Exon 1C mRNA in the knockout model (data not shown)."

Several pieces of raw data that were discussed in the main text have been added to the manuscript as figure supplements. In addition several pieces of new data have been added to the main figures in this revised manuscript. These include revisions to Figure 4B and the addition of Figures 7B, 7C, and 7E. In addition, the revised manuscript now contains no references to data that is not available to the reader.

- "Of particular significance, we discovered for the first time a novel CLU isoform that was localized to the mitochondrial matrix and was translated from Exons 3-9." This sentence in the Abstract is misleading, because after reading the article, it becomes apparent that the 45 kDa isoform has already been detected in human cell lines before, but was never characterized.

This sentence has been altered to correctly summarize the novelty of the manuscript. Now it reads “Moreover, we have characterized a 45 kDa mitochondrial CLU protein isoform (deemed mitoCLU) present in healthy human and rodent neurons, astrocytes and brain-derived cell lines. In addition, we show for the first time that mitoCLU is localized to the mitochondrial matrix of rodent brain and is translated from a non-canonical translational start site located at the second amino acid in Exon 3; a site that coincides with an in-frame methionine in human CLU.”

- Figure 1A: the authors claimed the lack of differences in the CLU protein expression profiles between male and females. However, the variance in some bands is striking, specifically in lungs and heart. It would be desirable to know if the authors analyzed these samples in the same gel in order to justify their conclusion. Moreover, as the authors remarked, LOAD more frequently impacts the female population, hence, it could be interesting to decipher whether the cellular localization of the different CLU isoforms vary between female and male brains, given that both sexes have similar protein expression profile in the brain. As claimed by published study from this group, clusterin could be a link between hypometabolic phenotype and increased amyloid dyshomeostasis in early onset of AD in female mice (Zhao et al., Neurobiology of aging, 2016).

The samples for male and female tissue panels were not run on the same gel, but were harvested and ran on the same day at the same time. Therefore no direct comparison of protein expression in any tissue was made between male and female mice. The purpose of this manuscript was to characterize CLU protein isoforms, however, we understand the importance of investigating this area.

With the knowledge that this manuscript contains several new avenues of CLU research, including sex differences at the subcellular level, which will be focused on in future investigations.

- Figure 1C. It is unclear which brain region (and consequently cell type) is analyzed in this immunohistochemistry experiment. Moreover, since the authors investigate the cell type-specific location of brain CLU proteins, I consider that a whole brain section should be tested with the same technique for this aim.

Several 4X images of the whole brain section that was utilized to generate the 40X images from Figure 1C were collected at the time of imaging. For reference, a 4X image of the brain showing MAP2 and DAPI immunostaining with a red box demonstrating a region of the dentate gyrus that was magnified in Figure 1C has been added in Figure 1—figure supplement 1. In addition, as these data are meant to demonstrate cellular expression of CLU, 40X GFAP/CLU staining from the same brain sections used in Figure 1C have also been added in Figure 1—figure supplement 1.

- The deglycosylation studies performed by the authors to claim that the proposed mitochondrial isoform of CLU is not glycosylated, consist in differences of molecular weight in protein immunoblots. However, some glycosylation patterns can be small, e.g. many O-linked glycosylation, not necessarily clear in a gel. Thus, I suggest to conduct a more robust experiment for post-translational modifications.

Studies investigating the possible O-linked glycosylation of CLU_45 kDa have been performed and are now included in the revised manuscript as the lower panel of Figure 7B. These data are consistent with our previous conclusions as no evidence of O-linked glycosylation is found.

- The authors declared that human CLU_45 kDa is translated from an AUG start site in Exon 3. This conclusion is solely based on overexpression of an Exon 3-9 construct in human SH-SY5Y cells. Here, I would suggest to refer to previously published evidence that this protein isoform is indeed expressed at basal levels in human cell lines and to show more evidence of this AUG codon being, indeed, the translation start site. There is the possibility that this protein isoform is also translated from a non-canonical start site, or that the translation is even initiated from another exon. The authors should perform a transfection experiment with a mutated AUG site to show that the AUG codon on Exon 3 is effectively the translation start site.

Data to support our previous assertion that “Human CLU_45 kDa is translated from an AUG start site in Exon 3” has been added as Figure 7C.

- "Moreover, as CLU_45 kDa is generated from a canonical translational start site in humans and is therefore likely translated at higher levels in the human brain, these data highlight the potential importance of mitochondrial CLU in human brain." This claim is overstating due to the lack of hard evidence about the nature of the translational start site and about the supposed higher level of translation in the human brain.

Agreed, this sentence has been removed from the text of the manuscript.

- "Moreover, we have identified a novel 45 kDa mitochondrial CLU protein isoform (deemed mitoCLU) present in healthy rodent brain tissue and both human and rodent brain-derived cells." "To our knowledge, this is the first publication to conclusively demonstrate the expression of a CLU protein isoform in healthy brain mitochondria (rodent or human)." The authors claim the expression of the mitoCLU isoform in human brains, while they only detected it in an overexpression study in human cell cultures.

Data indicating that CLU_45 kDa (mitoCLU) is expressed in human brain (astrocytes and neurons) have been added as Figure 7E in the revised manuscript.